# Nonlinearity of root trait relationships and the root economics spectrum

Deliang Kong [1,8], Junjian Wang [2,8], Huifang Wu [3,8], Oscar J. Valverde-Barrantes [4,8], Ruili Wang [5], Hui Zeng [6], Paul Kardol [7], Haiyan Zhang [1] & Yulong Feng [1]

The root economics spectrum (RES), a common hypothesis postulating a tradeoff between resource acquisition and conservation traits, is being challenged by conflicting relationships between root diameter, tissue density (RTD) and root nitrogen concentration (RN). Here, we analyze a global trait dataset of absorptive roots for over 800 plant species. For woody species (but not for non-woody species), we find nonlinear relationships between root diameter and RTD and RN, which stem from the allometric relationship between stele and cortical tissues. These nonlinear relationships explain how sampling bias from different ends of the nonlinear curves can result in conflicting trait relationships. Further, the shape of the relationships varies depending on evolutionary context and mycorrhizal affiliation. Importantly, the observed nonlinear trait relationships do not support the RES predictions. Allometry-based nonlinearity of root trait relationships improves our understanding of the ecology, physiology and evolution of absorptive roots.

[1] Liaoning Key Laboratory for Biological Invasions and Global Change, Shenyang Agricultural University, 110866 Shenyang, Liaoning Province, China. [2] School of Environmental Science and Engineering, Guangdong Provincial Key Laboratory of Soil and Groundwater Pollution Control, Southern University of Science and Technology, 518055 Shenzhen, China. [3] School of Life Sciences, Henan University, 475004 Kaifeng, China. [4] International Center of Tropical Botany, Florida International University, Miami, FL 33199, USA. [5] College of Forestry, Northwest A&F University, 712100 Yangling, China. [6] Key Laboratory for Urban Habitat Environmental Science and Technology, Peking University Shenzhen Graduate School, 518005 Shenzhen, China. [7] Department of Forest Ecology and Management, Swedish University of Agricultural Sciences, 90183 Umeå, Sweden. [8] These authors contributed equally: Deliang Kong, Junjian Wang, Huifang Wu, Oscar J. Valverde-Barrantes, Ruili Wang. Correspondence and requests for materials should be addressed to D.K. (email: deliangkong1999@126.com) or to J.W. (email: wangjj@sustech.edu.cn) or to Y.F. (email: yl_feng@tom.com)

Root foraging is essential for plant growth and ecosystem functioning. In most plants, the most distal and ephemeral portion of the root systems, referred to as absorptive roots, undertake this function[1,2]. Substantial interspecific trait variation among these absorptive roots has been reported for a variety of ecosystems and plant species pools[3]. One common hypothesis explaining this variation is that roots follow a leading dimension which reflects the acquisition-conservation tradeoff, i.e., the root economics spectrum (RES)[4,5]. Under the RES hypothesis, roots should follow a gradient in trait syndromes from fast foraging and short lifespan (i.e., acquisitive strategy) to slow foraging and long lifespan (i.e., conservative strategy). At global scales, the RES gradient has been used to understand root tissue function and in explaining responses of ecosystem carbon and nutrient cycling to climate change[6,7].

However, recent studies have found mixed support for some of the relationships predicted by the RES hypothesis. For instance, root tissue density (RTD), a trait frequently used as a key RES trait, should be positively correlated with root lifespan[8,9] and negatively correlated with root nitrogen concentration (RN), a proxy for nutrient acquisition rate[10]. Root diameter is another key trait depicting resource conservation and consistently shows a positive correlation with root lifespan[11,12]. Then, under the RES hypothesis, we would expect a positive correlation between root diameter and RTD and a negative correlation between root diameter and RN[11,13]. However, several studies have reported either no or a rather weak relationship (i.e., uncorrelated traits)[14–16], or even a significant negative relationship between root diameter and RTD and/or a positive relationship between root diameter and RN (i.e., correlated trait relationships)[17,18]. Both of these cases contradict predictions under the RES hypothesis. The relatively small species pools and/or the restricted geographic ranges considered in most previous studies limit the range of trait variation included, which could mask more universal trends. It is therefore important to test these trait relationships using a larger and global dataset[19,20].

Another possible reason for the contrasting findings may be that the trait relationships depart from the expected linear relationships assumed for the RES. Although nonlinear relationships prevail in biological processes due to the relationship between surface area and volume in functional organs[21,22], they have not been well recognized in plant roots. The two anatomical components of absorptive roots, namely, the stele tissue and the tissues outside the stele (ToS, including the epidermis, exodermis, and cortex), follow an allometric relationship; i.e., the thickness of the ToS (tToS) increases at a faster rate than the stele radius does from thin to thick absorptive roots[18,23,24]. As such, a nonlinear relationship exists between the proportion of root cross-sectional area occupied by the stele (PRS) and root diameter ($x$): $PRS = (1–2k-2cx^{-1})^2$, where $k$ and $c$ are parameters for the relationship between tToS and root diameter (i.e., $tToS = kx + c$, $c < 0$)[18]. Because the stele consists of lignified vascular tissue, it should be denser and have lower N-concentrations than the ToS; as such, the PRS should theoretically be positively correlated with RTD and negatively with RN[25,26]. Therefore, based on the above allometric relationship, we would expect a nonlinear negative relationship between RTD and root diameter, and a nonlinear positive relationship between RN and root diameter.

In addition to the RES, mycorrhizal affiliation is also an important factor shaping root systems and hence affecting root trait relationships. This is because different mycorrhizas (e.g., arbuscular (AM) versus ectomycorrhizal (EM) fungi) are typically associated with particular root morphologies and nutrient contents that are adapted to specific environmental conditions[14,15,27–29]. The mantle hyphae in EM species usually have low tissue density and have little correlation with RTD[16],

and therefore cannot explain the observed differences between EM and AM species. Instead, EM species typically dominate in nutrient-poor soils[28], which may lead to thicker and/or more intensely lignified root cell walls[30]. This, in turn, could potentially explain the higher RTD and lower RN in EM than in AM species[15,17,31,32]. In addition, we predict that the morphological modifications in cortex and/or stele tissues associated with the switch in mycorrhizal partnership would make inter-trait relationships for EM roots deviate from those predicted by the above-mentioned allometric relationship, and by that contribute to the observed variation in RES trait relationships. Additionally, studies on non-woody species tend to report stronger RES tradeoffs between root traits than studies on woody plants (e.g., Roumet et al.[33] for non-woody plants; Holdaway et al[17]. and Kong et al[18]. for woody plants). However, few studies have compared the two groups of species in the same context[30] with the same suit of root traits, making generalizations about the RES difficult. Testing relationships among RES traits across mycorrhizal types and between growth forms (i.e., woody and non-woody species) is thus instrumental for understanding plant strategies in resource acquisition and conservation as well as plant adaptation to different environments.

Here, we test the following two hypotheses: (1) Based on the allometric relationship between stele and cortical tissue in roots, relationships between RTD and RN, and root diameter should be nonlinear, and consequently do not follow the predictions based on the RES. More specifically, we expect similar nonlinear root trait relationships between woody and non-woody species as they both follow an allometric relationship between root stele and cortex[23]. (2) Nonlinear root trait relationships are weaker for EM than for AM plants as the harsher environments where EM plants grow would cause greater variation of RTD and RN by thickening and/or lignification of root cell walls. Testing nonlinear root trait relationships advances our understanding of the hypothesized RES, and could potentially reconcile the debate on this topic.

Here, using a global root trait dataset over 800 species, we find significant nonlinear relationships in woody but not in non-woody species between root diameter, and RTD and RN. Nonlinearity of the root trait relationships is attributed to the allometric relationship between root stele and cortical tissues and can explain how sampling bias from different ends of the nonlinear curves results in conflicting trait relationships observed in previous studies. Furthermore, the nonlinear relationships vary depending on evolutionary context and mycorrhizal affiliation, and do not support the RES predictions. Together, the allometry-based nonlinearity of root trait relationships greatly advances our understanding of the ecology, physiology and evolution of absorptive roots.

## Results

**Allometric relationships between root anatomical structures.** The root anatomical structures of both the woody and non-woody species followed allometric relationships for tToS (i.e., thickness of root tissues outside the stele) *vs.* root diameter and SR (i.e., stele radius) vs. root diameter, with regression slopes for woody species of 0.43 and 0.068 (linear regressions, $p < 0.001$) and regression slopes for non-woody species of 0.32 and 0.18 (linear regressions, $p < 0.001$), respectively (Fig. 1a; regression equations are presented in Supplementary Data 2). The regression slope of the relationship between tToS and root diameter was higher for woody than for non-woody species (standardized major axis, $p < 0.001$), while the regression slope between SR and root diameter was lower for woody than for non-woody species (standardized major axis, $p < 0.001$).

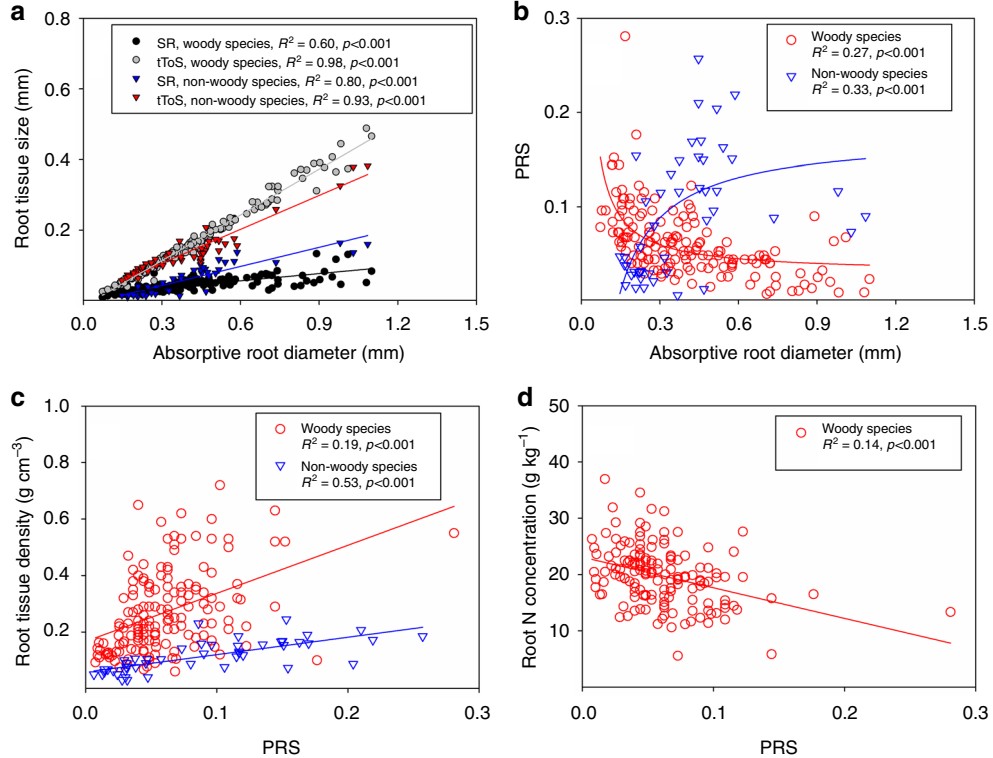

**Fig. 1** Allometry in absorptive roots. Allometric relationships between anatomical structures (**a**, **b**), and the relationships between the proportion of root cross-sectional area occupied by the steles (PRS) with root tissue density and root N concentration (**c**, **d**) for absorptive roots of woody and non-woody species. SR = stele radius, tToS = thickness of tissues outside the steles. See Supplementary Data 1 for the data sources, and Supplementary Data 2 for the regression equations. The insets show the fitness of the linear regressions (**a**, **c**, and **d**) and allometry-based nonlinear regressions (**b**)

For woody species, the PRS decreased with root diameter in a nonlinear way (nonlinear regression, $R^2 = 0.27$, $p < 0.001$), whereas for non-woody species, the PRS increased with root diameter in a nonlinear way (nonlinear regression, $R^2 = 0.33$, $p < 0.001$) (Fig. 1b). These contrasting relationships could be attributed to the negative intercept ($-0.016$) of the root diameter vs. tToS regression for woody species and the positive intercept (0.011) for non-woody species (Supplementary Data 2). The PRS was positively correlated with RTD in both woody (linear regression, $R^2 = 0.19$, $p < 0.001$) and non-woody species (linear regression, $R^2 = 0.53$, $p < 0.001$) (Fig. 1c), and it was negatively correlated with RN in woody species (linear regression, $R^2 = 0.14$, $p < 0.001$) (Fig. 1d).

**Nonlinear root trait relationships**. Across all species, RTD scaled negatively and nonlinearly with root diameter (nonlinear regression, $R^2 = 0.16$, $p < 0.001$), and the nonlinear relationship between RN and root diameter was rather weak and not significant (nonlinear regression, $R^2 = 0.002$, $p > 0.1$) (Fig. 2a, b; regression equations are presented in Supplementary Data 2). When accounting for effects of plant phylogeny for all species using phylogenetic independent contrasts (i.e., PICs), RTD was negatively and RN was positively correlated with root diameter, contradicting the expected RES trends (Supplementary Fig. 1a, b). Plant growth form (i.e., woody and non-woody species) and mycorrhizal type significantly affected the relationship between root diameter, RTD and RN across all species whether or not plant phylogeny was accounted for (except for the interaction effect of root diameter and growth form on RN in the phylogenetic generalized least squares analysis) (Supplementary Table 1; Fig. 2c–f, Supplementary Fig. 2).

When considered separately, the relationships of RTD and RN with root diameter were weak (Fig. 2e, f, Supplementary Fig. 1, 2; regression equations are presented in Supplementary Data 2) and unaffected by mycorrhizal type for non-woody species (Supplementary Table 2). In contrast, for woody species, the relationships of RTD and RN with root diameter differed among mycorrhizal types (Supplementary Table 3; Fig. 2c, d). RTD was negatively and nonlinearly correlated with root diameter when all woody species were analyzed together (nonlinear regression, $R^2 = 0.36$, $p < 0.001$) as well as within subsets of AM (nonlinear regression, $R^2 = 0.34$, $p < 0.001$), EM, (nonlinear regression, $R^2 = 0.39$, $p < 0.001$) and ERM (nonlinear regression, $R^2 = 0.74$, $p < 0.001$) woody species (Fig. 2c; regression equations are presented in Supplementary Data 2). The relationship between RN and root diameter was negative in EM woody species (nonlinear regression, $R^2 = 0.073$, $p < 0.001$) and positive in AM (nonlinear regression, $R^2 = 0.02$, $p < 0.001$) and ERM (nonlinear regression, $R^2 = 0.28$, $p < 0.001$) woody species (Fig. 2d; regression equations are presented in Supplementary Data 2). When accounting for effects of plant phylogeny using PICs, woody species showed a negative correlation between RTD and root diameter and a positive correlation between RN and root diameter (Supplementary Fig. 1c, d). For woody species, the nonlinear relationships between root diameter and RTD were similar when the 1st order roots and roots up to the 3rd order were analyzed separately (nonlinear regressions, $R^2 = 0.34$ and 0.42, $p < 0.001$) or together (nonlinear regression, $R^2 = 0.36$, $p < 0.001$), while the nonlinear relationship between root diameter and RN was better fitted when the two root order groups were analyzed separately (nonlinear regressions, $R^2 = 0.11$ and 0.18, $p < 0.001$) than when they were analyzed together (nonlinear regression, $R^2 = 0.035$, $p < 0.001$) (Supplementary Fig. 3; regression equations are presented in Supplementary Data 2).

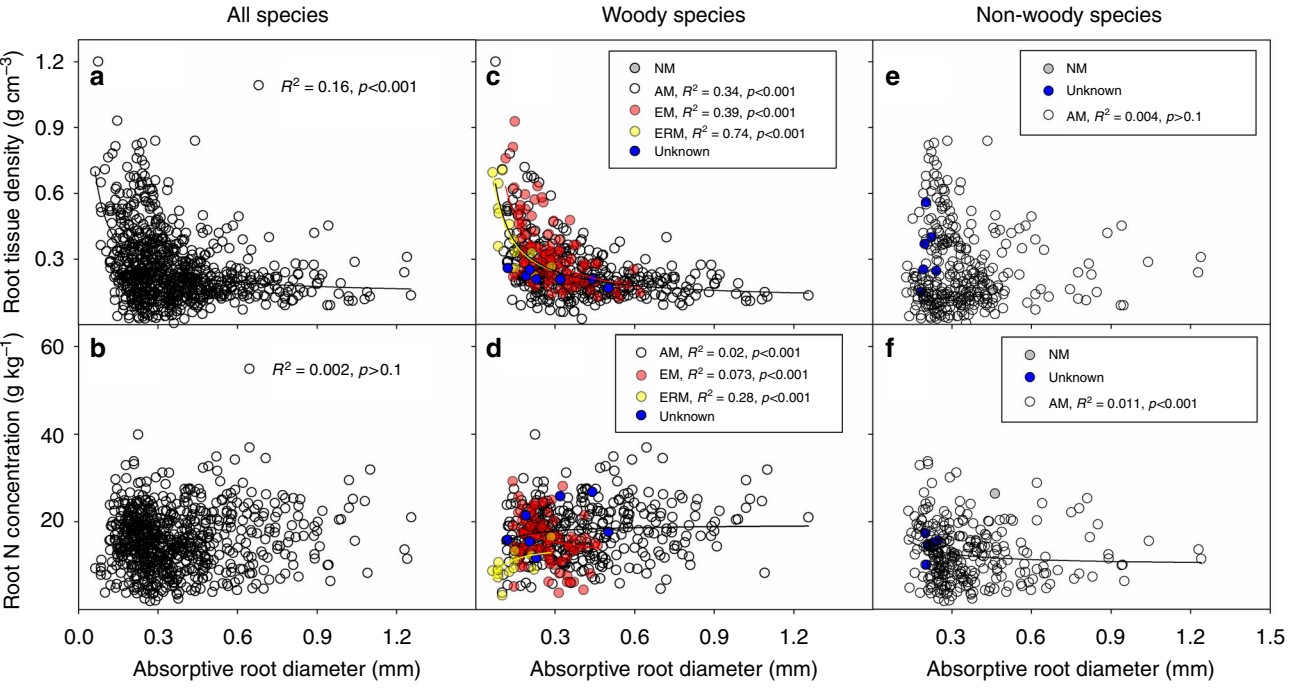

**Fig. 2** Nonlinear root trait relationships. Allometric root structure-derived nonlinear relationships between absorptive root diameter and root tissue density and root N concentration for different mycorrhizal types of all species (**a**, **b**), woody (**c**, **d**) and non-woody species (**e**, **f**). NM = non-mycorrhiza; AM = arbuscular mycorrhiza; EM = ectomycorrhiza; ERM = ericoid mycorrhiza; Unknown = unknown mycorrhizal status. The data sources are presented in Supplementary Data 1. For woody and non-woody species, the statistical significance for the regressions is only shown for the three main mycorrhizal types. The insets show the fitness of the allometry-based nonlinear regressions

**Table 1 Phylogenetic signals for root diameter, root tissue density (RTD) and root N concentration (RN)**

|  |  | Root diameter | RTD | RN |
|---|---|---|---|---|
| All species | Blomberg's K | **0.03** | 0.01 | **0.01** |
|  | Pagel's λ | **0.61** | **0.30** | **0.49** |
| Woody | Blomberg's K | **0.08** | 0.01 | 0.01 |
|  | Pagel's λ | **0.83** | **0.31** | **0.52** |
| Non-woody | Blomberg's K | 0.01 | 0.01 | **0.03** |
|  | Pagel's λ | <0.001 | **0.57** | **0.33** |

Bold values indicate significant phylogenetic signals ($p < 0.05$)

**Phylogenetic influence on root traits.** For all species, root diameter, RTD and RN were all influenced by plant phylogeny, and root diameter showed the strongest phylogenetic signal (Fig. 3; Table 1). For example, species in the clade of Magnoliids generally had thick roots, low RTD and high RN, whereas more recent lineages such as Rosids often showed the opposite pattern (Fig. 3). In woody species, root diameter showed the highest phylogenetic conservatism, while root diameter was little affected by phylogeny in non-woody species (Table 1).

**Effects of data source, root sampling, and climatic zone.** For woody species, the nonlinear relationships between RTD and root diameter and between RN and root diameter were better fitted for studies (i.e., data sources) reporting correlated trait relationships (i.e., negative RTD-root diameter and positive RN-root diameter correlations), $R^2 = 0.30$ and $R^2 = 0.049$) than for studies reporting uncorrelated traits ($R^2 = 0.06$ and $R^2 = 0.026$) (Fig. 4; regression equations are presented in Supplementary Data 2). The result of a linear mixed model for woody species showed that the data source (i.e., studies reporting correlated trait relationships vs.

studies reporting uncorrelated traits) had significant influence on the relationship between root diameter and RTD (i.e., root diameter × data source in the linear mixed model, $p < 0.001$, Supplementary Table 4). Root sampling (i.e., first order roots vs. roots up to the third order) did not affect the relationships of root diameter with RTD and RN (i.e., root diameter × root sampling in the linear mixed model, $p > 0.1$, Supplementary Table 4). Climatic zone did not affect RTD and RN (linear mixed model, $p > 0.05$, Supplementary Table 4). Finally, for woody species, studies reporting correlated trait relationships included a much larger proportion of species with high RTD and thin root diameter than studies reporting uncorrelated traits (53 vs. 17%) (Fig. 4a).

**Discussion**
Our global analysis of key root traits partially supports our first hypothesis of nonlinear relationships of RTD and RN with root diameter in woody (but not non-woody) species (Fig. 2, Supplementary Fig. 2). In woody species, the nonlinear relationships were similar or stronger in EM and ERM species that often exist in harsher environments than AM species;[28,34] this is inconsistent with our second hypothesis. This suggests that harsh environments may not necessarily exert a strong influence on cell wall thickening for EM and ERM roots in woody species. It is possible that the reduction of cortical tissue and evolutionary divergence of EM and ERM from their AM ancestors[27,35–37] (e.g., enzymes associated with these mycorrhizas for decomposition of plant litter or soil organic matter; the efficiency of carbon and nutrient interchange between fungi and roots in the symbioses) may enable EM and ERM species to adapt to harsher environments and conserve nonlinear trait relationships in EM and ERM roots. This, however, warrants further investigation.

Interestingly, for woody species we found a negative relationship between RN and root diameter in EM species, while the relationship was positive in AM and ERM species (Fig. 2d). The

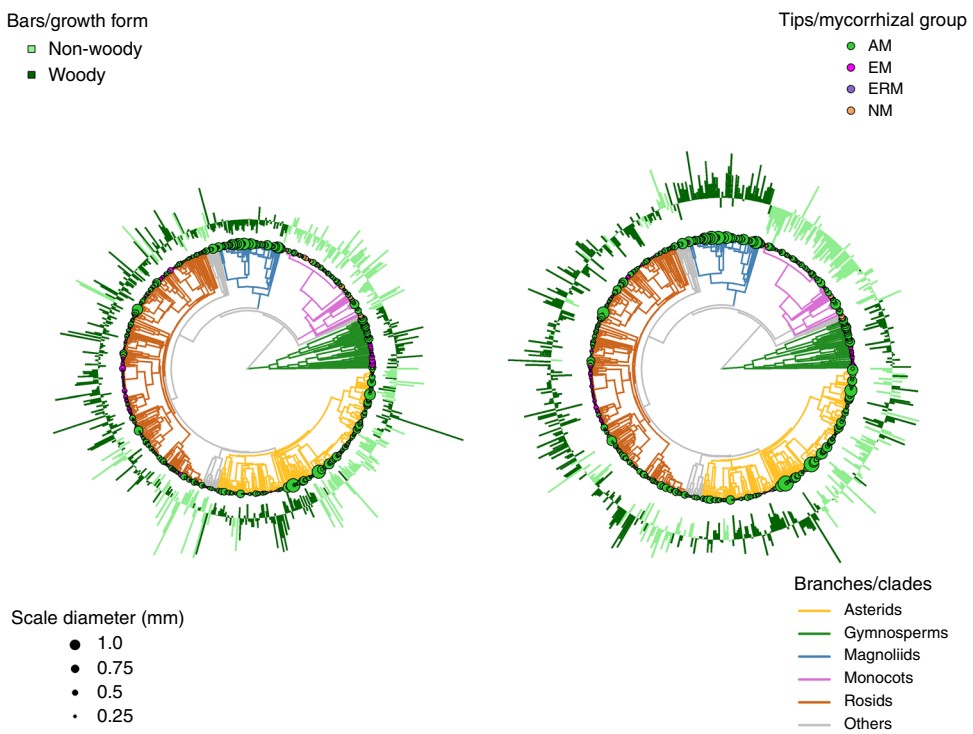

**Fig. 3** Variation of root traits with plant phylogeny. Variation in root diameter, root tissue density (RTD, left panel) and root N concentration (RN, right panel) with plant phylogeny, growth form and mycorrhizal type. The size of the circles at the tip of the phylogenetic tree is proportional to root diameter. Bars represent standardized values for RTD and RNC, with outward and inward bars representing values above and below the mean, respectively, and colors representing different growth forms. Branch colors in both phylogenies represent main phylogenetic clades and tips are proportional representations of root diameter colored by different mycorrhizal types: NM = non-mycorrhiza (orange); AM = arbuscular mycorrhizal (green); EM = ectomycorrhizal (magenta); ERM = ericoid mycorrhizal (violet)

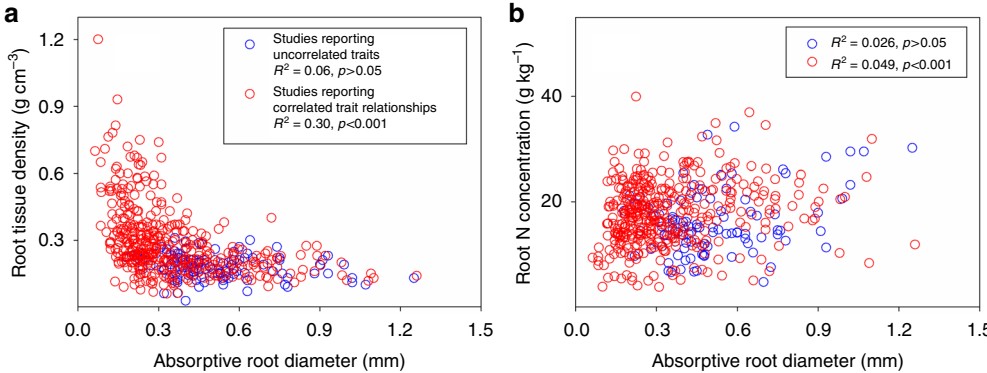

**Fig. 4** Root trait relationships for different data sources. Relationships between absorptive root diameter and root tissue density (RTD) (**a**) and root N concentration (RN) (**b**) in different sets of studies on woody species. Studies reporting correlated trait relationships (i.e., negative RTD- root diameter and positive RN-root diameter correlations) are indicated by red circles, and studies reporting uncorrelated trait relationships are indicated by blue circles. The insets show the fitness of the allometry-based nonlinear regressions

negative relationship in EM species could possibly be explained by two typical features of nutrient acquisition in EM species. First, for EM species, thin absorptive roots are covered by a relatively thick EM fungal mantle[16], which is relatively rich in N;[38] this thus enhances the root N concentration of thin roots compared to thick roots. This is notably different from AM species where thicker absorptive roots are usually associated with greater mycorrhizal colonization[18,24,39]. Second, EM species with thicker absorptive roots usually have less hyphal foraging precision (i.e., proliferation of extraradical hyphae in resource-rich patches)[40], which can reduce nutrient uptake and hence lower RN in thicker EM roots.

The nonlinear relationships in woody species can reconcile the current debate on the relationship between RTD and root diameter. We found weaker correlations of RTD and RN with root diameter for studies reporting no relationship between RTD and root diameter than for studies reporting negative relationships between RTD and root diameter (Fig. 4). The studies reporting negative relationships also included a higher proportion of species with thin roots and high RTD than studies reporting no relationships. Together, this demonstrates that those studies reporting no relationships focus on the region of slow decrease of RTD with increasing root diameter (Fig. 2c, Supplementary Fig. 2a). On the other hand, the negative relationship between RTD and

root diameter in studies reporting negative relationships may apply only to the part of the curve with a steep decrease. Therefore, nonlinearity of the root trait relationships could underpin how sampling bias from different parts of the nonlinear curves produces contradicting results as shown in recent studies. To gain a global picture of root trait relationships, it is therefore crucial to consider the full global range of trait variation across different lineages of woody species.

Recognition of the nonlinear relationship between RTD and root diameter could substantially improve our understanding of the tradeoffs among root functional traits. Theoretically, for an individual absorptive root (e.g., a single 1st order root), greater investment in dry mass could result in longer root lifespan following the cost-benefit theory[41,42] as observed for aboveground plant organs[19,22]. If we consider a plant root a cylinder formed by concentrically arranged tissues, dry mass of a single 1st order root must be a function of the total diameter as root volume increases exponentially with diameter (Supplementary Fig. 4). For the nonlinear relationship between RTD and root diameter (Fig. 2a, Supplementary 2c), if the increase of the root dry mass with increasing root diameter cannot be offset by the simultaneous decrease of tissue density, then root dry mass will always increase with root diameter. A modeling simulation based on the allometric relationship between root conductive and cortical tissue shows a monotonical increase of the root dry mass with increasing root diameter (Supplementary Fig. 4), suggesting a predominant role of root diameter rather than RTD in determining the root dry mass and hence root lifespan[22]. This is also supported by the consistent positive relationship between root lifespan and root diameter[12,43] and the lack of a relationship between root lifespan and RTD in woody species (Supplementary Fig. 5).

Furthermore, for AM species, root nutrient foraging activity may increase with increasing root diameter because of more resource allocation to non-cell wall fractions in the cortex (Supplementary Fig. 6), offsetting the relatively small changes in diameter of the lignified tissue in the stele (Fig. 1a)[18,24], and greater mycorrhizal colonization for thicker roots[12,27]. The higher nutrient foraging activity with increasing root diameter may have evolved to compensate for inefficient proliferation of thicker AM roots in resource rich patches[40,44–46]. Together, except for the negative relationship between root diameter and RN in EM species, the nonlinear trait relationships as revealed here suggest that root trait relationships do not necessarily align with the RES hypothesis. Therefore, the relative independence between changes in cortical and stele tissues in roots[24] indicates alternative acquisition strategies in plants mediated by their interaction with symbiotic fungi and further supports the multiple dimensionality in root trait syndromes proposed elsewhere[11].

Furthermore, the nonlinear relationship between root diameter and RTD advances our understanding of another key trait underlying the RES, i.e., specific root length (SRL, root length per unit root mass)[11,47]. Assuming that roots are cylindrically shaped, SRL can be expressed as: $SRL = 4/(\pi \times RTD \times root\ diameter^2)$[17,48]. If RTD is positively correlated with root diameter, as predicted by the RES, SRL then mathematically scales negatively with RTD. However, for the region of the nonlinear curve where RTD slowly decreases with root diameter (Fig. 2c), the negative effect of root diameter on SRL could counteract a potential positive effect of RTD on SRL, which, in turn, would lead to a positive relationship of SRL with RTD. In contrast, for the region of the nonlinear curve with fast decrease of RTD (Fig. 2c), SRL may show no relationship with RTD. This is because with increasing root diameter, the negative effect of root diameter on SRL could be offset by the potential positive effect of RTD on SRL. Together, the nonlinear relationship between RTD and root

diameter could explain an overall weak correlation of SRL with RTD, and also with RN (Supplementary Fig. 7a, b)[11,14,15,49] across the whole region of the nonlinear curve. Moreover, the weak correlation between SRL and RN and the strong coupling of SRL with root diameter[12,14,15,50] (also see Supplementary Fig. 7c) could also explain the relative weak correlation between root diameter and RN (Fig. 2). Together, these results illustrate how nonlinear root trait relationships can explain why SRL does not necessarily conform with the RTD-related plant economics spectrum in woody species[15,49].

Our results also suggest a phylogenetic component in the nonlinear root trait relationships, at least for woody species. Compared with the leveling off of RTD for thick roots in the above nonlinear relationships (Fig. 2), RTD decreases and RN increases continuously for thick roots even when excluding the influence of plant phylogeny (Supplementary Fig. 1c, d). The thick roots belong to early-derived angiosperms (e.g., Magnoliids, Fig. 3) associated with phosphorus-limited tropical soils, which suggests specialization of this root group to high dependence on AM fungi for nutrient foraging[12,25,29,51]. The weaker phylogenetic conservatism of RTD than root diameter (Table 1)[18,52] could probably be explained by specialization of early-derived angiosperm trees to phosphorus-limited soils by maximizing the cortex area for AM colonization, which would lead to low RTD and high RN. Later, during the evolution of novel angiosperm groups, natural selection promoted finer roots, either to constrain mycorrhizal colonization[12,18] or to optimize water conductivity and plant photosynthetic efficiency[23,29]. However, more research is needed to better understanding the different evolutionary mechanisms driving the transformation in root systems.

Different from woody species, we find no or rather weak nonlinear relationships between RTD, RN and root diameter in non-woody species, although both groups follow the same allometric relationships between stele and cortical tissue in absorptive roots. The allometric relationship has been proposed to optimize the balance between nutrient absorption via the cortex and transportation via conduits of the steles[23]. Compared with absorptive roots of woody species at a given diameter, non-woody species are reported to have about 30% less mycorrhizal colonization[12] while having a higher proportion of root cortex (t-test, $p < 0.01$) and thus lower RTD. This higher proportion of cortex in non-woody species might be associated with foraging strategies other than mycorrhizal colonization (e.g., metabolic activity). Therefore, in fertile soils, absorptive roots of non-woody species may be less dense and more active than absorptive roots of woody species[53] (Supplementary Fig. 6), and have more root hairs[54] and/or root branching[55]. Together, this might offset the lower nutrient acquisition through mycorrhizal associations in non-woody species. In contrast, in infertile soils, roots of woody species may not be much denser than non-woody species because higher RTD would reduce mycorrhizal colonization[15,16,18,27,56]. However, in infertile soils, roots of non-woody species could be denser[53] relative to woody species because of lower dependence on mycorrhizal colonization for non-woody than for woody species[12]. These speculations could partially explain why roots of non-woody species show greater variation of RTD and RN (and hence weak nonlinear root trait relationships) compared to woody species. Another reason for lack of nonlinear trait relationships in non-woody species may be that roots <2 mm in diameter include some non-absorptive roots[57] which typically have larger proportion of stele than absorptive roots[2], and as such, confound root trait relationships. In contrast, absorptive roots of woody species in previous studies (Supplementary Data 1) are sampled based on root branching order which can track the absorptive roots more precisely than the diameter-based method[1]. We therefore recommend for future studies, when

possible, to select absorptive roots based on branching order[1] rather than on root diameter.

In conclusion, our study demonstrates global nonlinear relationships of RTD and RN with root diameter among different mycorrhizal types of woody species but not for non-woody species. The differences between woody and non-woody species reveal insights into the ecological and evolutionary drivers of root structure in different plant life forms[58]. The nonlinear root trait relationships, a likely outcome of evolutionary constraints, could explain conflicting results among recent studies on the relationships of root diameter with RTD and RN. Interestingly, EM species show a different RN-root diameter relationship from that found in AM and ERM species, probably because EM species have a thinner fungal mantle and less hyphal foraging precision in thicker absorptive roots. Furthermore, our analyses show that the hypothesized RES[11] is not supported for absorptive roots of woody species, except for EM trees showing partial support of the RES. We advocate that the paradigm of nonlinearity for relationships between root diameter, RTD and RN provides a more rhizocentric way of viewing absorptive root ecological and physiological strategies.

## Methods

**Data collection**. We collected a global dataset of absorptive root traits (root diameter, RTD and RN) for 505 woody (455 angiosperms, 46 gymnosperms and four ferns) and 361 non-woody species from the Fine-Root Ecology Database (FRED)[3] as well as some other literature (see Supplementary Data 1 for details). In cases where root traits were presented only in figures, we extracted the trait values using SigmaScan Pro software (V5.0, SPSS Inc., Chicago, USA). For woody species, root traits were obtained from studies using either the first order (i.e., the most terminal root order) or up to the third orders of a root branch; in both cases, these are considered absorptive roots[1]. For non-woody species, we selected studies with fine roots less than 2 mm in diameter as most of these roots are absorptive[2,33]. When a trait of one species was reported in more than one study, we used the mean trait value across these studies as most of the multiple measurements of a species came from the same climatic zones (Supplementary Data 1). To establish relationships between root anatomy and root diameter, RTD and RN, we collected three root anatomical traits from the above species pools: stele radius (SR), tToS (thickness of root tissues outside the stele), and PRS (proportion of root cross-sectional area occupied by the stele). In total, there were 158 woody species with both the anatomical traits and RTD and RN for the firstorder roots, while these data were only available for 13 non-woody species (Supplementary Data 1).

In the dataset, we also recorded mycorrhizal status of the species which was obtained from published studies (Supplementary Data 1) as well as from databases of mycorrhizal classification[35,59]. For woody species, plants were classified into the following mycorrhizal types: AM (376 species), EM (89 species), ericoid mycorrhiza (ERM, 13 species), non-mycorrhiza (NM, three species), and dual mycorrhizas of AM and EM (AM + EM, 17 species). There were seven other woody species with unknown mycorrhizal status. For woody species, AM + EM were designated as EM based on their ability to diverge from its ancestral stage of AM[52]. Most of the non-woody species in our dataset were AM (347/361); the remaining species were NM (six species), unknown (three species) or dual mycorrhizas of AM and NM (AM + NM, five species). For non-woody species, AM + NM were considered as AM, because AM is more favored by natural selection than NM in these dual mycorrhizal associations[35].

**Data analyses**. Relationships between root diameter and stele radius (SR) and tToS were tested using linear regressions, separately for woody and non-woody species. We then tested the allometric relationship between the steles and the ToS by exploring whether the slopes are the same for the regression of SR with root diameter and the regression of tToS and root diameter. Differences in the allometric relationships between woody and non-woody species were analyzed by comparing slopes of the same regression between the two species groups. All slope differences were tested using standardized major axis (SMA) in SMATR version 2.0[60].

We used the following steps to test the nonlinear relationships of RTD and RN with root diameter as predicted from allometric root anatomical structures. First, we examined whether the PRS scaled with root diameter in the predicted nonlinear way (see Introduction for details). Then, we explored whether RTD and RN were linearly correlated with the PRS. For the woody species, two outlier data points were excluded, one with an exceptionally high RN (*Gironniera subaequalis*) and the other with an exceptionally high PRS (*Elaeocarpus hainanensis*). For the non-woody species, we did not test the relationship between the PRS and RN because the sample size for this relationship was too small (13 species, Supplementary Data 1). Finally, we tested whether RTD and RN each scale with root diameter in a

way similar to the relationship between the PRS and root diameter. The linear and nonlinear regressions were performed with the lm and nls function, respectively, in R software (v.3.30, R Core Team, 2016). This test was performed separately for woody and non-woody species, and for different mycorrhizal types. Additionally, for the woody species, these analyses were performed separately on the 1st order roots and roots up to the 3rd order as differences in the root diameter of these two categories of absorptive roots[57] may change the above trait relationships.

Given the phylogenetic conservatism of many root traits[18,52], we also explored the relationships between RTD and RN, and root diameter using phylogenetic independent contrasts (PICs)[61], which account for the influence of common ancestors on the trait relationships. PICs-based trait relationships were analyzed for the full dataset, as well as separately for woody species and non-woody species, respectively, using the package picante in R[62]. The plant phylogenetic tree was constructed according to an updated mega-phylogeny of vascular plants[63] and the Angiosperm Phylogeny website (http://www.mobot.org/MOBOT/research/APweb/). Species names in the phylogenetic tree were revised according to The Plant List (http://www.theplantlist.org/). The polytomies were resolved by the multi2di function in the ape package in R software. We estimated phylogenetic signals for root diameter, RTD and RN by employing Blomberg's K and Pagel's λ tests assuming a Brownian motion model of evolution[52]. We used phylogenetic generalized least squares and ordinary least squares[52,64] to test the effects of growth form (woody vs. non-woody) and mycorrhizal types (AM, EM, and ERM) on the relationships between root diameter and RTD and RN. We first conducted this analysis across all species and then separately for woody and non-woody species.

In woody species, we also tested whether the data source (i.e., studies reporting correlated trait relationships not supporting the RES: negative RTD-root diameter and positive RN-root diameter correlations vs. studies reporting uncorrelated traits) influenced the relationship between root diameter and RTD using a linear mixed model using the lmerTest package[65] in R. There were a few studies on woody species with low RTD reporting trait relationships only partially supporting the RES (e.g., both positive RTD-root diameter and RN-root diameter relationships, see Supplementary Data 1 for details). To test whether these few studies could influence the nonlinear root trait relationships for studies reporting correlated trait relationships, we tested the trait relationships both with and without these studies using the anova method in R. The influence of these studies was not significant ($F = 0.93$, $p = 0.54$ for the RTD-root diameter relationships; $F = 0.54$, $p = 0.93$ for the RN-root diameter relationships). Therefore, we added these studies to the studies reporting correlated trait relationships not supporting the RES. Then, using a linear mixed model, we tested the effects of the fixed factors (i.e., root diameter, data source, root sampling, climatic zone, and the study nested within the data source) and the interactions of root diameter with the data source and with root sampling, respectively. Climatic zones were classified as tropical, subtropical, temperate, boreal, and Mediterranean. Study (see Supplementary Data 1) was not considered as a random factor because they were not classified randomly but assigned to one of the data sources according to their trait relationships. Root sampling referred to studies collecting the first order roots and studies collecting roots up to the third order, respectively. Furthermore, for woody species, we explored whether the contradicting relationships between root diameter and RTD (see Introduction and Supplementary Data 1) as observed in previous studies were due to different species pools. To do so, we first compared the frequency of species with high RTD between studies reporting correlated trait relationships and studies reporting uncorrelated traits. The mean RTD ($0.236$ g cm$^{-3}$) across these studies was used to separate low- and high-RTD roots. We then ran nonlinear regressions for the relationships between RTD and root diameter and between RN and root diameter.

**Reporting summary**. Further information on research design is available in the Nature Research Reporting Summary linked to this article.

## Data availability

The data used in this study can be found in the studies listed in the Supplementary Data 1 and the Supplementary References affiliated to the Supplementary Figures, and are available upon request from the corresponding authors. The source data underlying the Supplementary Fig. 4c are provided as a Source Data file, and are deposited in the Dryad Digital Repository: https://doi.org/10.5061/dryad.sv2j191.

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

## Acknowledgements
We thank Jason Fridley, Katie M. Becklin, Mark E. Ritchie, Louis James Lamit, Alexander Richard Ebert, Joshua P. Schimel and the Plant Ecology Lab at Syracuse University, and Zeqing Ma for constructive comments on this study. This study was equally supported by funding from the National Key R&D Program of China (2017YFC1200101) and the National Natural Science Foundation of China (31670550, 31870522, 31470575 and 31670545). P.K. acknowledges support from the Swedish Research Council.

## Author contributions
D.K., J.W., H.W., H.Z. and Y.F. conceived the idea and collected the data, and R.W. and O.J.V-B. conducted the phylogenetic analyses. D.K., J.W., H.W., O.J.V., R.W., H.Z., P.K., H.Y.Z. and Y.F. discussed and contributed to the final framework of this study. D.K. wrote the first draft of the manuscript with significant help from P.K., J.W., O.J.V-B., H. W. and Y.F. All authors contributed to manuscript completion and revision.

## Additional information

**Competing interests:** The authors declare no competing interests.

