## [Peer Review File · Nature Communications]

Reviewer #1 (Remarks to the Author):

Dear Editor, Authors,

I read the manuscript by Dr. Kong and co-authors with great interest. By assuming linear relationships among root traits, previous studies have reported inconsistent results on the relationships between root tissue density, diameter and nitrogen (N) concentration. This manuscript shows 1) that root trait relationships between diameter, tissue density and N% are nonlinear due to the allometric scaling of anatomical structures with root diameter, and 2) how this confounds our insights in root trait relationships, particularly regarding a root economics spectrum. Given the current interest in root traits in terms of resource economics and the debate on whether such traits are organized in a global root economics spectrum (e.g. Ma et al. 2018, Nature) this is a novel, relevant and timely contribution to today's discussions in plant ecology.

I appreciate how the manuscript addresses some of the main methodological issues that have been raised to explain the lack of consistent RES relationships, i.e. the underlying phylogenetic properties of root traits, phylogenetic signals, the definition of absorptive roots and the effects of the species pool used. The approach and resulting outcomes are highly interesting.

My main concerns lie with the Discussion section. I am not sure whether I agree or disagree with the interpretation of the results, because this section is difficult to read and the argumentation hard to follow. I think the line of reasoning in the Discussion needs to be rewritten and perhaps restructured in order to be convincing – especially the results that do not agree with the hypotheses tested, e.g. the different trait relationships of woody and non-woody species, and the PIC analyses that lead to opposite relationships. I have listed larger and minor remarks in the attached Word document.

Yours sincerely,

Monique Weemstra

Reviewer #2 (Remarks to the Author):

In their manuscript "Non-linearity of root trait relationships and the root economics spectrum" the authors compiled a database of root diameter (RD), root tissue density (RTD) and root nitrogen concentration (RN) of absorptive roots of approximately 800 plant species. With this global fine root data they intended to resolve the so far mixed reports on the hypothesized tradeoff between resource acquisition and conservation traits according to the root economics spectrum (RES) and if

this differs for woody or non-woody species. In contrast to the predictions of the RES the analysis revealed a non-linear negative relationship between RD and RTD. The authors convincingly argue that this relationship stemmed from the allometric relationship between stele and cortical root tissue which they substantiate with trait information on stele radius (SR) and thickness of tissue outside the stele (tToS) on 158 woody species and 13 non-woody species. In addition, the authors test this non-linear relationship along an evolutionary context and suggest a phylogenetic component in it for woody species with weaker phylogenetic conservatism of RTD than RD. Moreover, their results suggest that ectomycorrhizal woody species show stronger nonlinear relationships than arbuscular mycorrhizal species.

The paper is very interesting and the research questions are timely and very much discussed in the community of root ecologists, functional ecologists and disciplines related to biodiversity research. I also believe the topic of the root economic spectrum and its potential dependence on additional factors is of interest to a wider scientific community. However, this perception is of course stained by personal interests. The study is well composed and the paper very well written. The database is remarkable and data analysis is relatively straight forward and appears to be appropriate although the description could be more detailed in places.

My major concern lies with the originality of the results provided in this study. Most of the ideas, hypotheses and results given in this paper have been shown elsewhere already. Admittedly, this was on more limited data or plant groups or geographic regions but still the major outcomes of the paper are not really new. Even the very concept itself – nonlinearity of the allometric relationship – was published before by the authors (Kong et al. 2017). Results on the phylogenetic relationship of RTD, RD and RN were recently published by Ma and colleagues for roughly 400 species (Ma et al. 2018) and the relationship of phylogenetic structured root traits and mycorrhizal colonization in Valverde-Barrantes (2016, 2017) . All these papers are correctly cited in the manuscript so I do not want to express here that the authors are not aware of this fact. Also, I am convinced that the more thorough dataset presented here fully warrants a new paper – I just think some of the wording could be better adjusted to the available knowledge.

The paper is based on a valuable database of root traits for more than 800 species. I am aware that the majority of this data is compiled from FRED yet substantial effort was put into additional data from other published literature. To my knowledge this database is not made publicly available or at least this is not stated in the manuscript or elsewhere in the accompanying material. I consider this fact a major flaw and would strongly encourage the authors to provide the compiled data e.g. to FRED for future use and ability of researchers to reproduce the work given the detail provided in this study.

Minor issues:

Line 53: “These nonlinear relationships explain how sampling bias from different ends of the nonlinear curves produces conflicting trait relationships.” I think this is a very important statement that is not really well highlighted in the rest of the paper. You do provide the data and kind of say this between the lines in the results and discussion, but I feel you could stress it even more that this is one potential cause for contradicting results in the field.

Line 73: I think reference 9 does not support the statement given here

Line 101: Space missing after point

Line 133: SR has not been defined before as abbreviation

Line 182 ff: I am not convinced your data and analysis does properly separate the climate and mycorrhizal type argument so I would recommend to step a bit more careful in the wording here. As shown by one of your authors (Valverde-Barrantes et al 2017 New Phytologist) climate has a stronger effect on trait variation than mycorrhizal type. In your analysis you average trait values per species for different data origin and thus the climatic signal should be much blurred.

Line 218 ff: You refer to a model presented in the supplement. If I get this correctly your line of argument here is a model shows RD is important for root dry mass. From this you conclude that RD is more important than RTD without further mentioning additional tests and also that root mass and lifespan are positively correlated without reference. So you conclude that there is a relationship between lifespan and RD which had already been shown - so why the model? I do not get the point here I am afraid.

Line 224: I thought root foraging capacity is higher with lower RD not increasing with RD?

Line 226: The background argument of higher RN in thicker roots is already in the intro and repeated here, yet you make it sound like a new argument. Perhaps cut this down in the intro than?

Line 229: which has been shown before. Sounds here like you show this for the first time.

Line 334: give more details on the linear mixed effect models.

Line 546: you say you use mean values per species for a trait (line 281) and that you have anatomical traits for 13 non-woody species. So why are there more than 13 points for non-woody species in these graphs? Do I miss something here?

Fig 4 shows exactly the same data as figure 2ab? Only the coloring differs. Could this be combined?

Responses to editor and reviewers

Note: All the line numbers in our response letter refer to the *revised* version.

Editor's comments

Your manuscript entitled "Nonlinearity of root trait relationships and the root economics spectrum" has now been seen by 2 referees. You will see from their comments below that while they find your work of interest, some important points are raised. We are interested in the possibility of publishing your study in Nature Communications, but would like to consider your response to these concerns in the form of a revised manuscript before we make a final decision on publication.

We therefore invite you to revise and resubmit your manuscript, taking into account the points raised. Please highlight all changes in the manuscript text file. We ask that you pay particular attention to the display and interpretation of your results, and provide further clarity in your argumentation.

RESPONSE: We sincerely thank the Editor for considering our manuscript. In the revised version, we have paid particular attention to presentation and interpretation of our results and we have simplified and clarified our reasoning and argumentation. Further, we have taken into account all reviewers' comments. Together, we believe, this has greatly improved our manuscript. Our detailed responses are listed below.

Reviewer 1

I read the manuscript by Dr. Kong and co-authors with great interest. By assuming linear relationships among root traits, previous studies have reported inconsistent results on the relationships between root tissue density, diameter and nitrogen (N) concentration. This manuscript shows 1) that root trait relationships between diameter, tissue density and N% are nonlinear due to the allometric scaling of anatomical structures with root diameter, and 2) how this confounds our insights in root trait relationships, particularly regarding a root economics spectrum. Given the current interest in root traits in terms of resource economics and the debate on whether such traits are organized in a global root economics spectrum (e.g.

Ma et al. 2018, Nature) this is a novel, relevant and timely contribution to today's discussions in plant ecology.

I appreciate how the manuscript addresses some of the main methodological issues that have been raised to explain the lack of consistent RES relationships, i.e. the underlying phylogenetic properties of root traits, phylogenetic signals, the definition of absorptive roots and the effects of the species pool used. The approach and resulting outcomes are highly interesting.

RESPONSE: We thank the reviewer for the positive evaluation and the constructive and helpful comments.

My main concerns lie with the Discussion section. I am not sure whether I agree or disagree with the interpretation of the results, because this section is difficult to read and the argumentation hard to follow. I think the line of reasoning in the Discussion needs to be rewritten and perhaps restructured in order to be convincing – especially the results that do not agree with the hypotheses tested, e.g. the different trait relationships of woody and non-woody species, and the PIC analyses that lead to opposite relationships. I have listed larger and minor remarks in the attached Word document.

RESPONSE: Based on the reviewer's comments, we have thoroughly reworked our manuscript, with particular attention to the line of reasoning in the Discussion section. For example, we have included a new paragraph emphasizing the implication of the nonlinearity of root trait relationships on the specific root length (SRL):

“Furthermore, the nonlinear relationship between root diameter and RTD advances our understanding of another key trait underlying the RES, i.e., specific root length (SRL, root length per unit root mass)^{11, 49}. Theoretically, SRL can be expressed as: $SRL = 4/(\pi \times RTD \times \text{root diameter} \times \text{root diameter})$ ^{17, 50}. If RTD is positively correlated with root diameter, as predicted by the RES, SRL then mathematically scales negatively with RTD. However, for the region of the nonlinear curve where RTD slowly decreases with root diameter (Fig. 2c), SRL could be positively related to RTD. This is because with increasing root diameter, the negative effect of root diameter could counteract the positive effect of RTD on SRL. In contrast, for the region of the nonlinear curve with fast decrease of RTD (Fig. 2c), SRL may show no relationship with RTD. This is because with increasing root diameter, the negative effect of root diameter on SRL could be offset by the positive effect of RTD on SRL. Together,

the nonlinear relationship between RTD and root diameter could explain an overall weak correlation of SRL with RTD, and also with RN (Supplementary Fig. 7a,b)^{11, 14, 15, 51}. The weak correlation between SRL and RN could also underlie a weak correlation between root diameter and RN given the wide demonstration of a strong correlation between root diameter and SRL^{12, 14, 15, 52} (also see Supplementary Fig. 7c). Together, these results illustrate how nonlinear root trait relationships can explain why SRL does not conform to the RTD-related plant economics spectrum in woody species^{15, 51}.” (Lines 271-288).

Further, we have reworked our argumentation for the weak nonlinearity in non-woody species by separately considering root strategies in rich and poor soil conditions:

“Compared with absorptive roots of woody species at a given diameter, non-woody species are reported to have about 30% less mycorrhizal colonization¹². In favorable (e.g., moist and/or fertile) soils, absorptive roots of non-woody species may be less dense³³ (Supplementary Fig. 6) with more root hairs⁵⁵ and/or root branching⁵⁶ than woody species; this would allow non-woody species to more actively take up nutrients balancing the lower nutrient acquisition through mycorrhizal associations. In contrast, in poor (e.g., dry and/or infertile) soils, roots of woody species may not be much denser than non-woody species because higher RTD would reduce mycorrhizal colonization^{18, 57} while woody species are in higher need of mycorrhizal association than non-woody species¹². However, in poor soils, roots of non-woody species may be denser with thinner stele vessels^{33, 58} relative to woody species as relative lower need of mycorrhizal colonization for non-woody species¹² especially for those species with finer absorptive roots⁵⁶. These could explain why roots of non-woody species show greater variation of RTD and RN than woody species at a given root diameter. This could potentially explain why we did not find nonlinear root trait relationships in non-woody species.” (Lines 307-321).

Please, see further our responses to the more detailed comments below.

Introduction

L. 81: Throughout the manuscript, the term ‘coupled relationships’ refers to a negative correlation between root diameter and tissue density, and a positive one between root N and diameter. This is confusing because coupled can also mean opposite trait correlations, especially in the context of a RES. It may be more clear to simply speak of positively, negatively or un-correlated traits? Also, were there any studies included that reported trait

correlations in line with the RES, and if so, these were not considered as ‘coupled’ data sources?

RESPONSE: We apologize for the lack of clarity. In line with the reviewer’s suggestion, we now use more straightforward terminology, i.e., positively correlated, negatively correlated or un-correlated traits throughout the manuscript.

A few examples:

“... whether the data source (i.e., studies reporting correlated trait relationships: negative RTD-root diameter and positive RN-root diameter correlations both non-supporting the RES vs. studies reporting un-correlated trait relationships)...” (Lines 408-410 in the Methods section).

“correlated trait relationships” and *“un-correlated trait relationships”* (Line 79, 81 in the Introduction section).

“studies reporting correlated trait relationships” and *“studies reporting un-correlated trait relationships”* (Lines 193-204 in the Result section).

“studies reporting no relationship between RTD and root diameter” and *“studies reporting negative relationships between RTD with root diameter”* (Lines 231-237 in the Discussion section, Lines 726-728 in the Figure legends, and Lines 765-767 in the caption of Fig. 4).

“Data source for studies on woody species: I: studies reporting correlated trait relationships, i.e., negative RTD-root diameter and positive RN-root diameter correlations, both not supporting the root economics spectrum (RES); II: studies reporting un-correlated trait relationships.” (Lines 15-16 for the revised footnote (d) of Supplementary Table 1).

Further, a few studies in our dataset reported trait relationships partially in line with the RES, e.g., a positive RTD-RD correlation in Xu (2011) and positive RTD-RD and RN-RD correlations in Valverde-Barrantes et al. (2015). To account for these studies in our analysis of the data source effect, we have revised the Methods and the Supporting information as follows:

“There were a few studies reporting above trait relationships only partially supporting the RES (e.g., both positive RTD-root diameter and RN-root diameter relationships, see Supplementary Table 1 for details) on woody species with low RTD. To test whether these few studies could influence the nonlinear root trait relationships for studies reporting the correlated trait relationships, we tested the trait relationships both with and without these studies using the anova method in R. The influence of these studies was not significant on the nonlinear trait relationships ($F=0.93$, $p=0.54$ for the RTD-root diameter relationships; $F=0.54$, $p=0.93$ for the RN-root diameter relationships). Therefore, we added these studies to the studies reporting correlated trait relationships.” (Lines 412-420 in the Methods section).

“There were a few studies for woody species under category I which partially supported the RES, i.e., reporting positive RTD-RD relationship in Xu (2011) and positive RTD-RD and RN-RD relationships in Valverde-Barrantes et al. (2015).”* (Lines 17-18 for the revised footnote (d) of Supplementary Table 1).

L. 98: Should ‘root diameter’ be replaced by ‘root tissue density’?

RESPONSE: We thank the reviewer for pointing out this mistake. We have corrected this in Line 98.

L. 101: Start a new paragraph here for the mycorrhizae

RESPONSE: Done (Line 102).

L. 105 – 111: I agree that trait relationships in the RES needs to take mycorrhiza into account, but how these relationships are affected by EcM roots needs more explanation. Both EcM associations and thicker cell walls have been associated with plants growing in poor environments (L. 105 – 109), but is there also a direct relationship (*Question 1*)? Do EcM roots have thicker cell walls (*Question 2*)? EcM and AM trees occur in environments that vary in soil fertility, so do EcM trees on richer soils still have denser tissue due to thick cell walls than AM trees on comparably poor soil (*Question 3*)? Based on this, EcM roots are hypothesized to be denser and lower in N% than AM roots (L. 107 – 108), for I assume a given diameter (*Question 4*)? I am curious about how the effect of fungal mantle on the diameter on EcM roots influences these different trait relationships between EcM and AM roots (*Question 5*)? If AM and EcM roots of the same diameter are being compared, the EcM roots would mostly include the fungal mantle (as most studies do not correct for this, but see

Withington et al. 2006 Ecol Monogr). As this is enriched in N compared to the roots (L. 196) EcM roots may have a higher N% than AM roots? Have the authors considered this effect(*Question 6*)? In order to better understand the hypotheses, the role of diameter needs to be clarified here.

RESPONSE: Thanks for these useful insights! Regarding *Questions 1 and 2*, we are not aware of any studies directly testing the relationship between EcM status and root cell wall thickness in poor environments. However, there is indeed some indirect evidence. For example, many studies have shown that EcM species usually have high RTD (Chen et al. 2013; Kramer-Walter et al. 2016; Comas et al. 2009). A recent study also showed that roots of some EcM species had higher root tissue density than roots of AM species for a given root diameter (Valverde-Barrantes et al. 2018). EcM hyphal mantle usually has much lower tissue density than root tissues and shows a very weak correlation with root tissue density ($r=0.14$, $p=0.73$ for the correlation of the mantle proportion in root cross sectional area with root tissue density; data from Withington et al. 2006). Therefore, the EcM hyphal mantle cannot explain the higher RTD of EcM species than AM species (for *Questions 4 and 5*). Instead, it is more likely that thickening and/or intensely lignifying root cell walls cause higher RTD for EcM than for AM species. In the revised manuscript, we have added some new discussion to address this point:

“As the mantle hyphae usually have low tissue density and have little correlation with RTD¹⁶, it is expected that for a given root diameter, higher RTD and lower RN in EM species than in AM species could result from thicker and/or more intensely lignified root cell walls in EM species^{15, 33, 34}” (Lines 109-112).

As for *Question 3*, it is possible that EcM species in fertile soils have higher RTD than AM species because EcM species have evolutionarily adapted to poor soils (e.g. Comas et al. 2012). For example, under greenhouse conditions with relatively fertile soils, it has been shown that EcM species can have higher RTD than AM species (Kramer-Walter et al. 2016). However, this idea warrants further investigation. In the revised manuscript, we have added two studies, i.e., Comas et al. 2012 (theoretical evidence) and Kramer-Walter et al. 2016 (indirect empirical evidence), to support our argument that *“This is because different mycorrhizas (e.g., arbuscular (AM) versus ectomycorrhizal (EM) fungi) are typically associated with particular root morphologies and nutrient contents that are adapted to specific environmental conditions^{14, 15, 27, 28, 29}.”* (Line 103-106).

As for *Question 6*, we did not consider the influence of the hyphal mantle on RN in EM species for two reasons. First, Compared with AM species, very few studies have explored the relationships between mycorrhizal fungal traits and RN in EcM species. Therefore, we are not convinced about the hypothesized relationship between root diameter and RN in EcM species. Second, the main aim in this section was to discuss the possibility of differences in root trait relationships between AM and EcM species based on differences in growth habitat (i.e., EcM species usually grow in harsher environments relative to AM species). Therefore, we prefer to discuss the possible role of the hyphal mantle in influencing RN in the Discussion section. Here, we have provided detailed argumentation for the hyphal mantle effect on RN, and we also refer to the indirect evidence of Withington et al. (2006):

“The negative relationship in EM species could possibly be explained by two typical features of nutrient acquisition in EM species. First, less root cross-sectional area is accounted for by the N-rich hyphal mantle⁴¹ with increasing root diameter of EM species ($r=-0.69$, $p=0.04$, data from Table 7 in Withington’s study¹⁶), whereas thicker absorptive roots in AM species can accommodate more AM fungi^{18, 27}. Second, EM species with thicker absorptive roots usually have less hyphal foraging precision (i.e., proliferation of extraradical hyphae in resource rich patches)⁴², which can reduce nutrient uptake and hence lower RN in thicker EM roots.” (Lines 221-228).

L. 123: ‘a’ (weaker nonlinear ...) needs to be removed (plural)

RESPONSE: Revised accordingly (Line 129).

L. 125: I agree this work would offer a relevant contribution to the debate on root traits, but it does not settle it – there are still many open questions; for instance, the RES is supposedly driven by the tradeoff between SRL and root lifespan which is not tested in this work, and the differences between woody and non-woody trait relationships are also far from clear.

RESPONSE: We agree that the word ‘settle’ was not appropriate and indeed a bit over-ambitious to cover these questions. We have replaced ‘settle’ with ‘reconcile’ (Line 133). We did not cover the tradeoff between SRL and root lifespan in our study ($r=-0.40$, $p=0.015$; figure not shown and very similar to other studies, e.g., Fig. 2a in Weemstra et al. (2016) and Extended Data Fig. 4c in Ma et al. (2018)). To integrate SRL into our argument of the nonlinearity of root trait relationships, we have added a new paragraph:

“Furthermore, the nonlinear relationship between root diameter and RTD advances our understanding of another key trait underlying the RES, i.e., specific root length (SRL, root length per unit root mass)^{11, 49}. Theoretically, SRL can be expressed as: $SRL = 4/(\pi \times RTD \times \text{root diameter} \times \text{root diameter})$ ^{17, 50}. If RTD is positively correlated with root diameter, as predicted by the RES, SRL then mathematically scales negatively with RTD. However, for the region of the nonlinear curve where RTD slowly decreases with root diameter (Fig. 2c), SRL could be positively related to RTD. This is because with increasing root diameter, the negative effect of root diameter could counteract the positive effect of RTD on SRL. In contrast, for the region of the nonlinear curve with fast decrease of RTD (Fig. 2c), SRL may show no relationship with RTD. This is because with increasing root diameter, the negative effect of root diameter on SRL could be offset by the positive effect of RTD on SRL. Together, the nonlinear relationship between RTD and root diameter could explain an overall weak correlation of SRL with RTD, and also with RN (Supplementary Fig. 7a,b)^{11, 14, 15, 51}. The weak correlation between SRL and RN could also underlie a weak correlation between root diameter and RN given the wide demonstration of a strong correlation between root diameter and SRL^{12, 14, 15, 52} (also see Supplementary Fig. 7c). Together, these results illustrate how nonlinear root trait relationships can explain why SRL does not conform to the RTD-related plant economics spectrum in woody species^{15, 51}.” (Lines 271-288).

As for the difference between woody and non-woody species, we discuss their differences in the Discussion section from the viewpoint of the nonlinearity of root trait relationships. However, we acknowledge that our previous argument was not very clear and could easily lead to confusing. We have therefore substantially reworked this section:

“Compared with absorptive roots of woody species at a given diameter, non-woody species are reported to have about 30% less mycorrhizal colonization¹². In favorable (e.g., moist and/or fertile) soils, absorptive roots of non-woody species may be less dense³³ (Supplementary Fig. 6) with more root hairs⁵⁵ and/or root branching⁵⁶ than woody species; this would allow non-woody species to more actively take up nutrients balancing the lower nutrient acquisition through mycorrhizal associations. In contrast, in poor (e.g., dry and/or infertile) soils, roots of woody species may not be much denser than non-woody species because higher RTD would reduce mycorrhizal colonization^{18, 57} while woody species are in higher need of mycorrhizal association than non-woody species¹². However, in poor soils, roots of non-woody species may be denser with thinner stele vessels^{33, 58} relative to woody species as relative lower need of mycorrhizal colonization for non-woody species¹² especially for those species with finer absorptive roots⁵⁶. These could explain why roots of non-woody

species show greater variation of RTD and RN than woody species at a given root diameter. This could potentially explain why we did not find nonlinear root trait relationships in non-woody species.” (Lines 307-321).

Furthermore, we have also considered the reviewer’s comments on the diameter cut-off of 2 mm as a potential factor explaining the weak nonlinearity of root trait relationships in non-woody species compared to woody species:

“Another reason for lack of nonlinear trait relationship in non-woody species may be that roots < 2mm in diameter include some non-absorptive roots⁵⁹ which typically have larger proportion of stele than absorptive roots², and as such, confound root trait relationships observed for non-woody species. However, absorptive roots of woody species in previous studies (Supplementary Table 1) are sampled based on root branching order which can track the absorptive roots more precisely than the diameter-based method¹. We therefore recommend for future studies to select absorptive roots based on branching order¹ rather than on root diameter.” (Lines 321-328).

Methods

L. 280 – 281 : At least for some species (e.g., grasses), still a large proportion of the roots < 2 mm may not be absorptive but transporting and it is considered a questionable threshold (Freschet & Roumet 2017 Funct Ecol). As transport roots are anatomically different from absorptive roots, e.g. by having a larger PRS (McCormack et al. 2015, New Phyt), this may confound the patterns observed for the non-woody species – depending on the species composition. The authors could mention this in their discussion.

RESPONSE: Thanks for this valuable comment. This indeed addresses an important challenge in across-taxa comparisons of root traits. We have included this point in our discussion:

“Another reason for lack of nonlinear trait relationship in non-woody species may be that roots < 2mm in diameter include some non-absorptive roots⁵⁹ which typically have larger proportion of stele than absorptive roots², and as such, confound root trait relationships observed for non-woody species. However, absorptive roots of woody species in previous studies (Supplementary Table 1) are sampled based on root branching order which can track the absorptive roots more precisely than the diameter-based method¹. We therefore

recommend for future studies to select absorptive roots based on branching order¹ rather than on root diameter.” (Lines 321-328).

L. 297 : ‘this’ should be ‘these’

RESPONSE: Revised accordingly (Line 368).

L. 315: ‘similar as’ should be ‘similar to’

RESPONSE: Revised accordingly (Line 387).

L. 322 – 323: I would like some more methodological information on the PIC analyses. To me it is not clear how to read Fig. S3: what do the x-axes represent, and which contrasts were used?

RESPONSE: We have added a some more details on the PIC analysis by referring to the classic paper by Felsenstein (1985), and providing more information on how we performed the analyses (the R package *picante*) and what these analyses account for:

“...using phylogenetic independent contrasts (PICs)⁶³, which account for the influence of common ancestors on the trait relationships. PICs-based trait relationships were analyzed for the full dataset, as well as separately for woody species and non-woody species, respectively, using the package picante in R⁶⁴. ” (Lines 393-397).

In Supplementary Fig. 1 (i.e., Fig. S3 in the initial submission), the x-axes represent contrasts of absorptive roots in total species (a,b), woody species (c,d) and non-woody species (e,f) after accounting for the influence of plant phylogeny. In the revised manuscript, we have added the specific species pools on which the contrasts are based below x-axes (i.e., Supplementary Fig. 1 in the new submission).

L. 334 – 341: Please mention that these analyses only included woody species (L. 7 Suppl. File), and whether studies that reported trait correlations in line with the RES were in- or excluded (or absent)?

RESPONSE: We now mention that these analyses were conducted for woody species only:

“In woody species, we also tested whether the data source...” (Line 408).

“...Furthermore, for woody species, we explored...” (Line 428).

Studies that reported trait correlations partially in line with the RES were included as follows:

“There were a few studies reporting above trait relationships only partially supporting the RES (e.g., both positive RTD-root diameter and RN-root diameter relationships, see Supplementary Table 1 for details) on woody species with low RTD.” (Lines 412-414 in the Methods section).

“There were a few studies for woody species under category I which partially supported the RES, i.e., reporting positive RTD-RD relationship in Xu (2011) and positive RTD-RD and RN-RD relationships in Valverde-Barrantes et al. (2015).”* (Lines 17-18 for the revised footnote (d) of Supplementary Table 1).

Results

There are many results, figures and tables, and different treatments, so a clear overview is needed. To this end, the Results section needs better structuring.

RESPONSE: We have restructured the Results section to make it read more smoothly. First, following the reviewer’s suggestion, we give a brief overview of the main results at the end of the Introduction.

“Using a global root trait dataset, we first tested the allometric relationships between root cortex and the stele for both woody and non-woody species. Then, we tested whether the allometric relationship could explain the nonlinearity of relationships of root diameter with RTD and RN for woody and non-woody species and for different mycorrhizal types. Finally, we evaluated how allometry-based nonlinearity may explain the conflicting data on root trait relationships observed in woody species and could reconcile the above debate.” (Lines 133-139).

Second, we have moved the paragraph on the role of phylogeny (i.e., Fig. 3 and Table 1, Lines 186-191) towards the end, after the results on growth forms (i.e., woody and non-woody) (Lines 167-185).

Here are some suggestions:

- The phylogenetic part (L. 143 – 146) needs to be presented elsewhere (later), because it breaks the flow of reading the other – more related – results, and is not directly related to the hypotheses (the figure numbers are also not in the right order).

RESPONSE: Thanks. We have moved the paragraph on the role of phylogeny (i.e., Fig. 3 and Table 1, Lines 186-191) towards the end, after the results on growth forms (i.e., woody and non-woody) (Lines 167-185).

- I prefer to see Fig S1a,b combined with Fig. 2, because it is the first result regarding the overall hypotheses (L. 147 – 149).

RESPONSE: Great suggestion! We have combined Fig. 2 and Supplementary Fig. 1a,b into one figure: the new Figure 2:

- As Figure S3 does not focus on mycorrhizal effects, L. 158 – 159 might fit better after L. 149, also because these PIC–correlations are negative for the species set as a whole. Add the direction of the relationships (positive and negative) when describing their significance (L. 147-148).

RESPONSE: Following the reviewer’s suggestion, we have moved the following sentence with the PIC results of all species after L. 149 of the initial submission:

“When accounting for effects of plant phylogeny for all species using phylogenetic independent contrasts (i.e., PICs), RTD was negatively and RN was positively correlated with root diameter (Supplementary Fig. 1a, b).” (Lines 159-162).

We have also added ‘negatively’ and ‘positive’ in this sentence to indicate the direction of the trait relationships:

“Across all species, RTD scaled negatively and nonlinearly with root diameter ($R^2=0.16$, $p<0.001$), and a rather weak positive and nonlinear relationship was found between RN and root diameter ($R^2=0.002$, $p<0.001$) (Fig. 2a, b, Supplementary Table 2).” (Lines 157-159).

- L. 167 – 169 could be moved to the beginning of the paragraph, so that the trait relationships within non-woody species are dealt with.

RESPONSE: Revised accordingly (see Lines 167-169).

L. 158 – 159: Again, the PIC results need some more explanation; what am I looking at in Figure S3?

RESPONSE: We have revised this sentence as follows:

“When accounting for effects of plant phylogeny using PICs, woody species showed a negative correlation between RTD and root diameter and a positive correlation between RN and root diameter (Supplementary Fig. 1c, d).” (Lines 174-177; please, note that Supplementary Figs. 1c,d correspond with Figs. S3c,d in our initial submission).

L. 170 – 176, and Fig. 4: The authors need to specify that these analyses only included woody species

RESPONSE: Revised accordingly (see below).

“For woody species, the nonlinear relationships...” (Line 192).

“...The results of linear mixed model for woody species showed that...” (Lines 195-196).

“...Finally, for woody species...” (Line 202).

“...in different sets of studies on woody species...” (Line 765).

Figures and Tables: I prefer not having to look up statistics in supplementary or main text, so I suggest to:

- indicate significance in Figures 1, 2 and 4, or in the caption;

RESPONSE: We have added statistical information to Figures 1, 2 and 4:

Fig. 1

Fig. 2

Fig. 4

- add significance levels (with asterisks) in the supplementary tables;

RESPONSE: Revised accordingly.

- add legends in supplementary figures (especially as the same colors are being used to show different factors in different graphs).

RESPONSE: We have added legends to Supplementary Figs. 1, 2 and 3:

Supplementary Fig. 1

Supplementary Fig. 2

Supplementary Fig. 3

Figure 3 nicely summarizes the phylogenetic results, but it is not very clear in print (when you cannot zoom in); perhaps the authors can

- Change the legends, as they are confusing. For instance, change the legend referring to the clades into lines, and the legend referring to growth form into bars instead of dots.

- Use more discernible colors for at least ERM and EM.
- Perhaps add a legend on the size of the circles that represent absorptive root diameter (e.g. 3 size classes with the corresponding root diameter value) to get a general idea of the values

RESPONSE: We have revised the figure according to these suggestions. In the revised Fig. 3, the left panel looks slightly smaller as the species pool included was smaller than in the right panel.

Fig. 3

Discussion

The authors claim their 1st hypothesis was supported by their analyses (L. 179 – 180), but this was only the case for the woody species, i.e. 60 % of all species, and even among them, relationships between root N% and diameter were rather weak (L. 148, $R^2 = 0.002$). The results are still very interesting and the nonlinearity of these traits is an important advance in understanding root trait (co-)variation, but the results not/little in line with the hypotheses deserve more attention. Moreover, some of the explanations regarding the results need to be clarified or better supported. As the paper places its results in the context of a RES,

relationships with other traits could be more extensively discussed, and it has interesting results to do so.

RESPONSE: In the revised version, we have replaced “support” by “partially support”.

“Our global analysis of key root traits partially supports our first hypothesis...” (Line 208).

For sections not/little in line with the hypotheses, we have reworked the text to more accurately reflect our findings. For example, we have thoroughly revised our discussion of the relationship between RN and root diameter in EcM species which did not support our hypothesis (Lines 220-228).

Further, we have clarified the Results by using clearer language and more straightforward argumentation. For example, we have clarified the explanation for the weaker nonlinear root trait relationship in non-woody species than in woody species, which was confusing in our initial submission (Lines 307-321).

We have also added more discussion on the implications of nonlinear root trait relationship for another key root trait, i.e., SRL (Lines 271-288).

Please, see below for our more detailed responses to these comments.

L. 184: use ‘exist’ rather than ‘existed’; unless this was actually tested?

RESPONSE: Done (Line 211).

L. 186 – 187: I don’t understand this sentence ‘Thus, relationships’. I don’t see what it refers to, and how it relates to growth habit: does growth habit refer to growth form (woody vs non-woody), or something else? The authors should rephrase or explain this.

RESPONSE: We apologize for being unclear. Growth habitat (‘habit’ was not the correct word) refers to the relatively harsher environments for EM and ERM species compared with AM species. Specifically, we initially expected that the nonlinear root trait relationships (e.g. RD-RTD) could be weaker in EM species compared with AM species because EM species usually grow in harsher environments than AM species. For a given root diameter, RTD could be increased in harsher environments by thickening and/or lignifying root cell walls, which

could cause a deviation of the RTD from the predicted value based on the allometric relationship (i.e., Fig. 1a in the main text). Therefore, the harsher environments for EM species could result in a weakened RD-RTD relationship for EM roots compared with AM roots. However, our results showed similar RD-RTD relationships between EM and AM species, which did not support these expectations. Therefore, our results suggest that the harsh environments where EM species grow may not exert a strong influence on cell wall thickening as to significantly affect the RD-RTD relationship for EM roots. We have revised this sentence as follows by using ‘harsh environments’ instead of “habitat”:

“This suggests that harsh environments may not necessarily exert a strong influence on cell wall thickening for EM and ERM roots in woody species.” (Lines 212-214).

L. 194 – 197: Please, rephrase or clarify the explanation as to why thick roots of EcM species have lower N% than thin roots (as opposed to Am/ErM species). First, the authors state it is due to ‘... a lower proportion of roots covered by the hyphal mantle ...’. I guess they mean that thicker EcM roots have lower cover than thin EcM roots, and therefore lower N%? The second explanation refers to the lower nutrient foraging by EMM with increasing diameter. I assume the authors imply that thicker roots of EM species have less foraging precision, less nutrient uptake, and hence, lower root N (based on the literature referred to)? In any case, this needs more and clear explanation in order to interpret the results.

RESPONSE: Yes, the two reasons we used to explain the negative root diameter and RN relationship in EcM species are in line with the reviewer’s understanding. To further clarify these points, we have rewritten these sentences and added more detailed information on how lower RN in thicker EcM roots can be explained:

“The negative relationship in EM species could possibly be explained by two typical features of nutrient acquisition in EM species. First, less root cross-sectional area is accounted for by the N-rich hyphal mantle⁴¹ with increasing root diameter of EM species ($r=-0.69$, $p=0.04$, data from Table 7 in Withington’s study¹⁶), whereas thicker absorptive roots in AM species can accommodate more AM fungi^{18, 27}. Second, EM species with thicker absorptive roots usually have less hyphal foraging precision (i.e., proliferation of extraradical hyphae in resource rich patches)⁴², which can reduce nutrient uptake and hence lower RN in thicker EM roots.” (Lines 221-228).

L. 199 Remove ‘together’; this is redundant in combination with ‘pooling’.

RESPONSE: Done (Line 231).

L. 212 – 213, and L. 220 - 221: The ‘greater investment of absorptive roots in dry mass’ confuses me because greater mass investment can also imply more root length (i.e. at the root system level). From the following lines, the authors seem to refer to greater mass investments per unit root length?

RESPONSE: Yes. At the root system level, greater investment in dry mass of absorptive roots can result from: 1) thinner root diameter, greater total root length and thus less root dry mass per unit root length; 2) thicker root diameter, longer single root length (because of the positive correlation between root length and root diameter, Chen et al. 2013; Kong et al. 2014) but less total root length and thus higher root dry mass per unit root length. In the first case, roots may have a shorter lifespan because of their thinner diameter (see Supplementary Fig. 5 and also Ma et al. 2018 Nature), which contradicts our argument of higher root mass resulting in longer root lifespan according to the cost-benefit theory (see Line 246). Therefore, our argument is indeed based on the individual root level, e.g., a single 1st order root. We have clarified this as follows:

“Theoretically, for an individual absorptive root (e.g., a single 1st order root), greater investment in dry mass could result in longer root lifespan following the cost-benefit theory⁴³,⁴⁴ as applied to aboveground plant organs^{19, 22}.” (Lines 245-247).

Our argument following the above sentence for model in Supplementary Fig. 4 is based on root dry mass instead of root dry mass per unit root length. This is because our argument of “*greater investment of absorptive roots in dry mass could result in longer root lifespan* (Lines 245-246)” is based on the positive dry mass-lifespan relationship of plant organs (Marbà et al. 2007) as well as the well-recognized cost-benefit theory (Eissenstat et al. 2000). We have clarified this point as follows:

“If we consider a plant root a cylinder formed by concentrically arranged tissues, dry mass of a single 1st order root must be a function of the total diameter as root volume increases exponentially with diameter (Supplementary Fig. 4).” (Lines 247-249).

L. 210 – 232: The part about other root trait tradeoffs w.r.t. the RES can be further completed:
L. 220 – 221: The authors state that root diameter rather than RTD influences root mass and hence lifespan. Just to verify, this claim is derived from the fact that root mass still increases with root diameter (Fig. S4), even though RTD decreases with diameter (which would decrease root mass)? As SRL is considered a key trait in the RES, I suggest that the authors discuss the consequences of the nonlinearity of root traits for SRL. The RES hypothesis assumes diameter and RTD to be positively correlated to each other and negatively to SRL. But if they are not positively related – as shown here – it may explain why RTD is often found uncorrelated to SRL, and why neither SRL and diameter consistently correlate to root N% (Weemstra et al. 2016 New Phyt).

RESPONSE: Yes, our claim that “*a predominant role of root diameter rather than RTD in determining the root dry mass (Lines 255-256)*” is indeed based on theoretical and empirical data (see Supplementary Fig. 4a,b). According to the allometric relationship between root cortex and stele, RTD decreases with increasing root diameter. Therefore, with increasing root diameter, root dry mass can be positively affected by root diameter but negatively affected by RTD. To determine which of the two is more important in determining root mass, we made a model as shown in Supplementary Fig. 4a. This model shows that the negative effect of RTD on root mass can be offset by the positive effect of root diameter on root mass, thus causing root mass to still increase with root diameter. The Supplementary Fig. 4b is an indirect evidence supporting our model in Supplementary Fig. 4a by using data of root diameter and PRS (i.e., proportion of root cross-sectional area occupied by the steles).

To further validate our ideas on the positive relationship between root diameter and root dry mass, we have used a dataset of the single 1st order root mass (see the Source Data file in this submission) and root diameter of 96 woody species from our previous study (Kong et al. 2014). The high goodness-of-fit of the cubic regression between the two traits directly and strongly supports the dominant role of root diameter in determining root mass (Supplementary Fig. 4c).

Supplementary Fig. 4

We very much appreciate the comments on our results in relation to SRL. In the revised version, we have added a new paragraph articulating the consequences of the nonlinearity of root traits for SRL:

“Furthermore, the nonlinear relationship between root diameter and RTD advances our understanding of another key trait underlying the RES, i.e., specific root length (SRL, root length per unit root mass)^{11, 49}. Theoretically, SRL can be expressed as: $SRL = 4/(\pi \times RTD \times \text{root diameter} \times \text{root diameter})$ ^{17, 50}. If RTD is positively correlated with root diameter, as predicted by the RES, SRL then mathematically scales negatively with RTD. However, for the region of the nonlinear curve where RTD slowly decreases with root diameter (Fig. 2c), SRL could be positively related to RTD. This is because with increasing root diameter, the negative effect of root diameter could counteract the positive effect of RTD on SRL. In contrast, for the region of the nonlinear curve with fast decrease of RTD (Fig. 2c), SRL may show no relationship with RTD. This is because with increasing root diameter, the negative effect of root diameter on SRL could be offset by the positive effect of RTD on SRL. Together, the nonlinear relationship between RTD and root diameter could explain an overall weak correlation of SRL with RTD, and also with RN (Supplementary Fig. 7a,b)^{11, 14, 15, 51}. The weak correlation between SRL and RN could also underlie a weak correlation between root diameter and RN given the wide demonstration of a strong correlation between root diameter and SRL^{12, 14, 15, 52} (also see Supplementary Fig. 7c). Together, these results illustrate how nonlinear root trait relationships can explain why SRL does not conform to the RTD-related plant economics spectrum in woody species^{15, 51}.” (Lines 271-288).

L. 223 – 225: Nutrient foraging is suggested to increase with diameter due to a larger cortex relative to stele size. Yet thicker roots are expected to grow slower, reducing their competitive ability to resources from a rich soil patch compared thin-rooted species (Eissenstat 1992, J Plant Nutr; Comas et al. 2014, Frontiers Plant Sci). This other side of the nutrient uptake abilities of thicker roots should be mentioned too.

RESPONSE: We agree with the reviewer and have added this aspect of nutrient foraging for thick roots:

“The higher nutrient foraging activity with increasing root diameter may have evolved to compensate for inefficient proliferation of thicker AM roots in resource rich patches^{42, 46, 47, 48}” (Lines 263-264).

L. 229: For EM species, root diameter and RTD were also negatively correlated (L. 157). Please check this.

RESPONSE: Yes, root diameter and RTD were negatively correlated. This is not in line with the RES. To clarify this point, we have revised this sentence as follows:

“Together, except for the negative relationship between root diameter and RN in EM species, the nonlinear trait relationships as revealed here suggest that root trait relationships do not necessarily align with the RES hypothesis.” (Lines 265-267).

L. 237: Replace ‘are usually’ by e.g. ‘belong to’

RESPONSE: Revised accordingly (Line 293).

L. 248: Replace ‘despite’ by ‘although’

RESPONSE: Revised accordingly (Line 304).

L. 251 – 260: I disagree with this explanation, but I am not sure if I fully understand the line of reasoning. If I misunderstood, this paragraph needs to be reformulated because it is difficult to follow. - The authors propose that root traits relationships differ for non-woody species because they depend less on mycorrhiza and more on other strategies than woody species. Yet the studies referred to do not compare mycorrhizal dependency of woody vs non-woody species. Branching may be an important strategy for trees too (Kong et al. 2014 New Phyt) and is also influenced by mycorrhization (Liese et al. 2017 Frontiers Plant Sci), and mycorrhization is important for many non-woody species. The claim that nutrient foraging strategies are fundamentally different seems an overstatement, and does not sufficiently

explain these differences between woody vs nonwoody species (L. 260). Could the diameter cut-off point of 2 mm contribute to this results?

RESPONSE: We acknowledge that we did not compare the mycorrhizal colonization of absorptive roots between woody and non-woody species. In the revised version, we have revised this sentence by referring to a recent meta-analysis by Ma et al. (2018), which shows on average 30% less mycorrhizal colonization for non-woody species than for woody species (Lines 307-309). However, we agree with the reviewer that mycorrhizal colonization is also an important nutrient foraging strategy in non-woody species. Recent studies show that mycorrhizal colonization in non-woody species is related to some root traits, e.g., negatively correlated with root branching intensity and positively correlated with root diameter (see Li et al. 2017). Therefore, the lower mycorrhizal colonization in non-woody species could be associated with higher root branching intensity for nutrient foraging, especially for non-woody species with finer roots. In the revised manuscript, we have restructured the argumentation concerning the mycorrhizal association in non-woody species. Specifically, we introduce two cases (i.e., rich vs. poor soils) that could contribute to the greater variation in RTD and hence weak or lack of nonlinear root trait relationships in non-woody species:

“Compared with absorptive roots of woody species at a given diameter, non-woody species are reported to have about 30% less mycorrhizal colonization¹². In favorable (e.g., moist and/or fertile) soils, absorptive roots of non-woody species may be less dense³³ (Supplementary Fig. 6) with more root hairs⁵⁵ and/or root branching⁵⁶ than woody species; this would allow non-woody species to more actively take up nutrients balancing the lower nutrient acquisition through mycorrhizal associations. In contrast, in poor (e.g., dry and/or infertile) soils, roots of woody species may not be much denser than non-woody species because higher RTD would reduce mycorrhizal colonization^{18, 57} while woody species are in higher need of mycorrhizal association than non-woody species¹². However, in poor soils, roots of non-woody species may be denser with thinner stele vessels^{33, 58} relative to woody species as relative lower need of mycorrhizal colonization for non-woody species¹² especially for those species with finer absorptive roots⁵⁶. These could explain why roots of non-woody species show greater variation of RTD and RN than woody species at a given root diameter. This could potentially explain why we did not find nonlinear root trait relationships in non-woody species.” (Lines 307-321).

In line with the reviewer’s suggestion, we have added another possible explanation for the patterns observed in non-woody species (i.e., the root diameter cut-off point of 2 mm):

“Another reason for lack of nonlinear trait relationship in non-woody species may be that roots < 2mm in diameter include some non-absorptive roots⁵⁹ which typically have larger proportion of stele than absorptive roots², and as such, confound root trait relationships observed for non-woody species. However, absorptive roots of woody species in previous studies (Supplementary Table 1) are sampled based on root branching order which can track the absorptive roots more precisely than the diameter-based method¹. We therefore recommend for future studies to select absorptive roots based on branching order¹ rather than on root diameter.” (Lines 321-328).

- If root hairs and root branching have a large impact on the balance between cortex and stele, I suspect it would cause variation both between and within woody and non-woody species. And does this imply that the cortex : stele ratio would be lower for non-woody than woody species (that presumably depend less on mycorrhiza, but still need to transport their resources)?

RESPONSE: We have revised our argumentation for root hairs and root branching with regarding to different mycorrhizal colonization between woody and non-woody species (see the following example):

“Compared with absorptive roots of woody species at a given diameter, non-woody species are reported to have about 30% less mycorrhizal colonization¹². In favorable (e.g., moist and/or fertile) soils, absorptive roots of non-woody species may be less dense³³ (Supplementary Fig. 6) with more root hairs⁵⁵ and/or root branching⁵⁶ than woody species; this would allow non-woody species to more actively take up nutrients balancing the lower nutrient acquisition through mycorrhizal associations” (Lines 307-312).

Different from the expectation of the reviewer, we find that non-woody species have a lower stele:root diameter ratio than woody species (2.4 vs. 2.8, $p < 0.01$). This suggests that non-woody species have a higher proportion of cortex than woody species. The higher proportion of cortex in non-woody could be related to other functions (e.g., more root hairs and greater root branching, Schweiger et al. 1995; Li et al. 2017) rather than being associated with the lower mycorrhizal colonization in non-woody species.

- Moreover, the effect of variation of RTD in response to changing soil conditions in mycorrhizal species on trait relationships has earlier been discarded (L. 185 – 186). Why would it apply here?

RESPONSE: Actually, the argument for L. 185-186 in our initial submission referred to woody species. We have corrected this:

“This suggests that harsh environments may not necessarily exert a strong influence on cell wall thickening for EM and ERM roots in woody species.” (Line 212-214).

- The reference to Fig. S6 (L. 256) should be placed after ‘thickening root cell walls’, because the relationship with the soil is not demonstrated here.

RESPONSE: We have rewritten the argumentation in this section (307-318).

L. 256: ‘... in response to ...’

RESPONSE: We have rewritten the argumentation in this section (307-318).

L. 268: As also brought up above, I do not understand why the RES does hold for EM trees; it is only confirmed for the relationship between diameter and N%, but not with RTD (L. 155 – 158).

RESPONSE: The reviewer is correct that these trait relationships only partially support the RES for EM species. Therefore, we have revised this sentence as follows:

“..., except for EM trees showing partial support of the RES” (Line 336).

Reviewer 2

In their manuscript “Non-linearity of root trait relationships and the root economics spectrum” the authors compiled a database of root diameter (RD), root tissue density (RTD) and root nitrogen concentration (RN) of absorptive roots of approximately 800 plant species. With this global fine root data they intended to resolve the so far mixed reports on the hypothesized tradeoff between resource acquisition and conservation traits according to the root economics spectrum (RES) and if this differs for woody or non-woody species. In contrast to the

predictions of the RES the analysis revealed a non-linear negative relationship between RD and RTD. The authors convincingly argue that this relationship stemmed from the allometric relationship between stele and cortical root tissue which they substantiate with trait information on stele radius (SR) and thickness of tissue outside the stele (tToS) on 158 woody species and 13 non-woody species. In addition, the authors test this non-linear relationship along an evolutionary context and suggest a phylogenetic component in it for woody species with weaker phylogenetic conservatism of RTD than RD. Moreover, their results suggest that ectomycorrhizal woody species show stronger nonlinear relationships than arbuscular mycorrhizal species.

The paper is very interesting and the research questions are timely and very much discussed in the community of root ecologists, functional ecologists and disciplines related to biodiversity research. I also believe the topic of the root economic spectrum and its potential dependence on additional factors is of interest to a wider scientific community. However, this perception is of course stained by personal interests. The study is well composed and the paper very well written. The database is remarkable and data analysis is relatively straight forward and appears to be appropriate although the description could be more detailed in places.

RESPONSE: We greatly appreciate these positive comments, and we are glad to hear that the reviewer thinks our study is very interesting.

My major concern lies with the originality of the results provided in this study. Most of the ideas, hypotheses and results given in this paper have been shown elsewhere already. Admittedly, this was on more limited data or plant groups or geographic regions but still the major outcomes of the paper are not really new. Even the very concept itself – nonlinearity of the allometric relationship – was published before by the authors (Kong et al. 2017). Results on the phylogenetic relationship of RTD, RD and RN were recently published by Ma and colleagues for roughly 400 species (Ma et al. 2018) and the relationship of phylogenetic structured root traits and mycorrhizal colonization in Valverde-Barrantes (2016, 2017) . All these papers are correctly cited in the manuscript so I do not want to express here that the authors are not aware of this fact. Also, I am convinced that the more thorough dataset presented here fully warrants a new paper – I just think some of the wording could be better adjusted to the available knowledge.

RESPONSE: Thanks for these thoughtful comments. Our study builds on previous work (including our own), but the current manuscript is novel for the following reasons:

1) In Kong et al. (2017) we just presented a hypothesis, i.e., ‘*the nutrient absorption-transportation balance*’, to explain the allometric relationship between root cortex and stele which was first demonstrated in our previous study on absorptive roots (Kong et al. 2014). However, our current manuscript uses a unique, very large global dataset to demonstrate the nonlinearity of key root trait relationships based on allometric relationships (Fig. 1). Furthermore, in our current study, we show that the nonlinearity can explain conflicting results among recent studies on root trait relationships and could potentially reconcile the emerging debates on the RES. This is a big step forward in our global understanding of the ecology, physiology and evolution of absorptive roots.

2) Moreover, our current study addresses several other important points that were neither considered in our previous work nor in other studies. First, differences in the allometric relationships between woody and non-woody species. Our current study shows that both woody and non-woody species follow allometric relationships between root cortex and stele, while the allometries are different (Fig. 1a) and have different consequences for the nonlinearity of root trait relationships for woody and non-woody species (Fig. 2). Second, differences in the nonlinearity of root trait relationships among different mycorrhizal types. We show that the root diameter vs. RTD relationship is overall similar among AM, EM and ERM species, while the root diameter vs. RN relationship is dramatically different between AM and EM species (see Lines 220-228). Third, the role of phylogeny. Based on the phylogenetic analyses, we demonstrate a phylogenetic component in the nonlinear root trait relationships (Fig. 3, Supplementary Fig. 1; Supplementary Table 3-5; also see Lines 289-302).

3) The relationships between root diameter and RTD and RN were explored by Ma et al. (2018). However, Ma et al. (2018) only referred to the positive relationship between root diameter and RTD and, importantly, did not considering these relationships within the context of nonlinearity. Furthermore, the allometry of root cortex and stele, an important factor contributing to nonlinearity, was not covered in Ma et al. (2018).

4) The studies by Valverde-Barrantes et al. helped building the framework of our current work, but these studies did not cover the key topics of our current manuscript, i.e., the allometry-based nonlinearity of root trait relationships and difference in nonlinearity of root trait relationships among mycorrhizal types and different plant growth forms (i.e., woody and non-woody). Specifically, Valverde-Barrantes et al. (2015 Functional Ecology) explored the

phylogenetically-structured root morphological traits, but this study *only included AM species*). Valverde-Barrantes et al. (2016 Plant and Soil) tested the effects of plant phylogeny on arbuscular mycorrhizal colonization, but this study did not cover the nonlinearity of root trait relationships. Valverde-Barrantes et al. (2017 New Phytologist) tested the effects of climate, phylogeny, growth form and mycorrhizal type on root traits and the relationships between root and leaf traits, but this study did not explore root trait relationships.

The paper is based on a valuable database of root traits for more than 800 species. I am aware that the majority of this data is compiled from FRED yet substantial effort was put into additional data from other published literature. To my knowledge this database is not made publicly available or at least this is not stated in the manuscript or elsewhere in the accompanying material. I consider this fact a major flaw and would strongly encourage the authors to provide the compiled data e.g. to FRED for future use and ability of researchers to reproduce the work given the detail provided in this study.

RESPONSE: Indeed, a large part of the dataset used in our study can be found in FRED. For those studies not included in FRED, we provide detailed information on how to access the data (see Supplementary Table 1, footnote (c)):

1) One dataset by Wang et al. (2008) is not available in FRED, but the data are open and available at <https://datadryad.org/resource/doi:10.5061/dryad.3rj81>, or upon request to Dr. Wang, R.L. (wangrl@nwafu.edu.cn).

2) Two other datasets also not available in FRED: Xu (2011) (see Table 2-1, Fig. 3-2, Fig. 3-8 and Fig. 5-2) and Jia 2011 (see Table 2.1, Fig. 3.1, Fig. 3.5 and Fig. 3.10); a MSc (thesis in Chinese with figures and tables also shown in English). These trait data are not available in FRED possibly because they will be used for the researchers' own studies. However, these two studies can be openly accessed at <http://cdmd.cnki.com.cn/Article/CDMD-10225-1011146348.htm> and at <http://cdmd.cnki.com.cn/Article/CDMD-10232-1011198993.htm>, and the trait values can easily be extracted. Data for these two studies are also available upon request to the corresponding authors for reproducing our work (deliangkong1999@126.com; yl_feng@tom.com).

Below, we give two examples of the data presented in Xu (2011) (left) and Jia (2011) (right):

图 3-1 古田山 50 个树种前 5 级根的平均直径，每个树种各根序上不同小写字母表示差异显著 ($P < 0.05$)

Fig. 3.1 Mean diameter for the first five order roots of 50 species in Gutianshan, lowercase letters indicate significant ($P < 0.05$) differences among individual root orders of each species.

图 3-2 中国热带 27 个阔叶树种前 5 级根的平均直径，每个树种各根序上不同的小写字母表明差异显著 ($P < 0.05$)

Fig. 3-2 Mean diameter for the first five order roots of twenty-seven tropical hardwood species in China. Lowercase letters within a species indicate significant ($P < 0.05$) differences among individual root orders

Minor issues:

Line 53: “These nonlinear relationships explain how sampling bias from different ends of the nonlinear curves produces conflicting trait relationships.” I think this is a very important statement that is not really well highlighted in the rest of the paper. You do provide the data and kind of say this between the lines in the results and discussion, but I feel you could stress it even more that this is one potential cause for contradicting results in the field.

RESPONSE: Accepted. We have emphasized this point in the Discussion and conclusion section:

“Therefore, nonlinearity of the root trait relationships could underpin how sampling bias from different parts of the nonlinear curves produces contradicting results as shown in recent studies.” (Lines 238-240).

“The nonlinear root trait relationships, a likely outcome of evolutionary constraints, could explain conflicting results among recent studies on the relationships of root diameter with RTD and RN.” (Lines 332-334).

Line 73: I think reference 9 does not support the statement given here

RESPONSE: We have replaced this reference with a suitable reference (Ryser et al. 1996 Functional Ecology) (Line 73).

Line 101: Space missing after point

RESPONSE: It should be a new paragraph and we have corrected this mistake (Line 102).

Line 133: SR has not been defined before as abbreviation

RESPONSE: We apologize for the mistake. SR means stele radius, and we have now indicated this in Line 143.

Line 182 ff: I am not convinced your data and analysis does properly separate the climate and mycorrhizal type argument so I would recommend to step a bit more careful in the wording here. As shown by one of your authors (Valverde-Barrantes et al 2017 New Phytologist) climate has a stronger effect on trait variation than mycorrhizal type. In your analysis you average trait values per species for different data origin and thus the climatic signal should be much blurred.

RESPONSE: Yes, climate can affect root traits (see Valverde-Barrantes et al. 2017; Freschet et al. 2017). In our study, most of the ‘average trait values’ are from woody species. We have carefully examined the woody species and found that most of the species with average root trait values (86 of 89 species) were measured in the same climatic zones or in subtropical and tropical zones with very similar mean annual precipitation and temperature. Therefore, climate may not greatly affect the average trait values of a species in different studies. To clarity, we have revised the Methods section as follows:

“... , we used the mean trait value across these studies as most of the multiple measurements for a species came from the same climatic zones (Supplementary Table 1).” (Lines 351-353).

Moreover, in the revised manuscript, we also account for the climate effect on trait variation by using climatic zone as a fixed actor in the linear mixed model analysis, and the climate effect is not significant:

“...we tested the effects of the fixed factors (i.e., root diameter, data source, root sampling, climatic zone, and the study nested within the data source) and the interactions of root diameter with the data source and with root sampling, respectively.” (Lines 421-423).

“...Climatic zone did not affect RTD and RN ($p>0.05$, Supplementary Table 6).” (Lines 201-202).

Line 218 ff: You refer to a model presented in the supplement. If I get this correctly your line of argument here is a model shows RD is important for root dry mass. From this you conclude that RD is more important than RTD without further mentioning additional tests and also that root mass and lifespan are positively correlated without reference. So you conclude that there is a relationship between lifespan and RD which had already been shown - so why the model? I do not get the point here I am afraid.

RESPONSE: We apologize for being unclear and the confusion this may have caused. Yes, our model (Supplementary Fig. 4a, see below) is used to demonstrate the dominant role of RD in root dry mass. Actually, Supplementary Fig. 4b provides indirect evidence for testing the role of RD in root dry mass, for which we use data of RD and PRS (i.e., proportion of root cross-sectional area occupied by the stele) presented in Supplementary Table 1. However, as the reviewer points out, we did not provide “additional tests”, i.e., tests using root mass and root diameter data. To address this issue, in the revised version, we do provide such a direct test, i.e., Supplementary Fig. 4c where we use the data of 1st order root mass (see the Source Data file in this submission) and root diameter of the 96 woody species from one of our previous studies (Kong et al. 2014). The new Supplementary Fig. 4c shows that root mass indeed increases with increasing root diameter, further supporting our argument for the dominant role of root diameter in determining root mass.

Supplementary Fig. 4

Therefore, this model is important as it establishes a connection between root diameter and root dry mass. Given that plant organ mass is usually a proxy for lifespan (see the following paragraph), our model as such can provide a possible explanation for why root diameter is positively related to root lifespan (i.e., *via* the effect of root diameter on root dry mass).

We acknowledge that we lack direct evidence of a positive correlation between root mass and root lifespan. However, we have some indirect evidence for this argument. First, roots with greater mass but shorter lifespan will be naturally selected against according to the cost-benefit theory (Eissenstat et al. 2000). Second, plant lifespan has been demonstrated to be in a 1/4 power relationship with plant mass across wide range of species ‘from tiniest phototrophs to the largest trees’ (Marbà et al. 2007). Therefore, although empirical tests are urgently needed, it is reasonable to assume a positive relationship between root mass and root lifespan in absorptive roots. To clarify this point, we have revised this section by referring to the ‘cost-benefit theory’ and added references (i.e., Marbà et al. 2007, Eissenstat and Yanai 1997, and Eissenstat et al. 2000) as indirect evidence for the positive relationship between the mass and lifespan of plant organs:

“Theoretically, for an individual absorptive root (e.g., a single 1st order root), greater investment in dry mass could result in longer root lifespan following the cost-benefit theory⁴³,⁴⁴ as applied to aboveground plant organs^{19, 22}.” (Lines 245-247).

Line 224: I thought root foraging capacity is higher with lower RD not increasing with RD?

RESPONSE: Our argument here is based on the single root level rather than the root system level (see also our responses to comment of Reviewer 1 to L. 212-213 and L. 220-221 in our initial submission). At the root system level, lower RD may be associated with a greater total root length and hence a greater root foraging capacity. However, at the individual root level (e.g., a single 1st order root), thicker RD in AM species is usually associated with greater mycorrhizal colonization (Baylis 1975; Kong et al. 2014) and a higher RN because of a lower stele and cell wall proportion (Fig. 1b, Supplementary Fig. 7), both of which indicate higher foraging activity. We have added this clarification to the Results:

“Theoretically, for an individual absorptive root (e.g., a single 1st order root)...” (Line 245).

In the revised manuscript, we also acknowledge some disadvantage of thicker absorptive roots, i.e., the lower root proliferation in resource rich patches, as also pointed out by the Reviewer 1:

“The higher nutrient foraging activity with increasing root diameter may have evolved to compensate for inefficient proliferation of thicker AM roots in resource rich patches^{42, 46, 47, 48}” (Lines 263-264).

Line 226: The background argument of higher RN in thicker roots is already in the intro and repeated here, yet you make it sound like a new argument. Perhaps cut this down in the intro than?

RESPONSE: We prefer to keep this argument here because it follows closely from the argument for *“the relative independence between changes in cortical and stele tissues in roots”* (Lines 267-268), which indicates much greater variation in the cortex than in the stele with increasing root diameter across species (see Fig. 1a and Valverde-Barrantes et al, 2016).

Line 229: which has been shown before. Sounds here like you show this for the first time.

RESPONSE: We have changed this sentence as follows:

“Together, except for the negative relationship between root diameter and RN in EM species, the nonlinear trait relationships as revealed here suggest that root trait relationships do not necessarily align with the RES hypothesis.” (Lines 265-267).

Line 334: give more details on the linear mixed effect models

RESPONSE: We have added more details on the linear mixed model by explicitly specifying the fixed and interaction effects:

“Then, using a linear mixed model, we tested the effects of the fixed factors (i.e., root diameter, data source, root sampling, climatic zone, and the study nested within the data source) and the interactions of root diameter with the data source and with root sampling, respectively. Climatic zones were classified as tropical, subtropical, temperate, boreal, and mediterranean. Study (see Supplementary Table 1) was not considered as a random factor because they were not classified randomly but assigned to one of the data sources according

to their trait relationships. Root sampling referred to studies collecting the 1st order roots and studies collecting roots up to the 3rd order, respectively.” (see Lines 420-428).

Line 546: you say you use mean values per species for a trait (line 281) and that you have anatomical traits for 13 non-woody species. So why are there more than 13 points for non-woody species in these graphs? Do I miss something here?

RESPONSE: In Fig. 1a,b,c the number of species is much greater than 13 (see also Supplementary Table 2). However, there are only 13 species for which both root anatomy and RN have been reported; therefore, we did not show the PRS-RN relationship in Fig. 1d. To avoid any potential confusion, we have added “for this relationship” after “because the sample size”. (Line 385).

Fig 4 shows exactly the same data as figure 2ab? Only the coloring differs. Could this be combined?

RESPONSE: We have considered this idea. However, as one of the key points in this study is to show how the large global dataset can reconcile different inter-trait relationships among previous studies (with which the Reviewer 1 also agrees), we feel that it is helpful for the reader to keep Fig. 4 separate from Fig. 2b. The data for Figs. 2 and 4 are almost the same, except that in Fig 2b all values for the same plant species were averaged across all studies, while in Fig. 4 values for the same plant species were averaged within each of the two categories (i.e., ‘studies reporting correlated trait relationships vs. studies reporting un-correlated trait relationships’).

References cited in the response letter

- Baylis GTS 1975.** The magnolioid mycorrhiza and mycotrophy in root systems derived from it. In: Sanders FE, Mosse B, Tinker PB eds. *Endomycorrhizas*. London: Academic Press, 373–389.
- Chen W, Koide RT, Adams TS, DeForest JL, Cheng L, Eissenstat DM. 2016.** Root morphology and mycorrhizal symbioses together shape nutrient foraging strategies of temperate trees. *Proceedings of the National Academy of Sciences USA* **113**(31): 8741-8746.
- Chen W, Zeng H, Eissenstat DM, Guo D. 2013.** Variation of first-order root traits across climatic gradients and evolutionary trends in geological time. *Global Ecology and Biogeography* **22**(7): 846-856.

- Comas LH, Eissenstat DM. 2009.** Patterns in root trait variation among 25 co-existing North American forest species. *New Phytologist* **182**(4): 919-928.
- Comas LH, Mueller KE, Taylor LL, Midford PE, Callahan HS, Beerling DJ. 2012.** Evolutionary patterns and biogeochemical significance of angiosperm root traits. *International Journal of Plant Sciences* **173**(6): 584-595.
- Eissenstat DM, Wells CE, Yanai RD, Whitbeck JL. 2000.** Building roots in a changing environment: implications for root longevity. *New Phytologist* **147**(1): 33-42.
- Eissenstat DM, Yanai RD 1997.** The ecology of root lifespan. In: Begon M, Fitter AH eds. *Advances in Ecological Research*: Academic Press, 1-60.
- Felsenstein J. 1985.** Phylogenies and the comparative method. *American Naturalist* **125**: 1-15.
- Freschet GT, Valverde-Barrantes OJ, Tucker CM, Craine JM, McCormack ML, Violle C, Fort F, Blackwood CB, Urban-Mead KR, Iversen CM, et al. 2017.** Climate, soil and plant functional types as drivers of global fine-root trait variation. *Journal of Ecology* **105**(5): 1182-1196.
- Jia Q. 2011.** *Functional traits of fine roots and their relationship with leaf traits of 50 major species in a subtropical forest in Gutianshan*. Master, Qiqihar University (China).
- Kong D, Ma C, Zhang Q, Li L, Chen X, Zeng H, Guo D. 2014.** Leading dimensions in absorptive root trait variation across 96 subtropical forest species. *New Phytologist* **203**(3): 863-872.
- Kong D, Wang J, Zeng H, Liu M, Miao Y, Wu H, Kardol P. 2017.** The nutrient absorption–transportation hypothesis: optimizing structural traits in absorptive roots. *New Phytologist* **213**(4): 1569-1572.
- Kramer-Walter KR, Bellingham PJ, Millar TR, Smissen RD, Richardson SJ, Laughlin DC, Mommer L. 2016.** Root traits are multidimensional: specific root length is independent from root tissue density and the plant economic spectrum. *Journal of Ecology* **104**(5): 1299-1310.
- Li H, Liu B, McCormack ML, Ma Z, Guo D. 2017.** Diverse belowground resource strategies underlie plant species coexistence and spatial distribution in three grasslands along a precipitation gradient. *New Phytol* **216**(4): 1140-1150.
- Ma Z, Guo D, Xu X, Lu M, Bardgett RD, Eissenstat DM, McCormack ML, Hedin LO. 2018.** Evolutionary history resolves global organization of root functional traits. *Nature* **555**(7694): 94-97.
- Marbà N, Duarte CM, Agusti S. 2007.** Allometric scaling of plant life history. *Proceedings of the National Academy of Sciences USA* **104**(40): 15777-15780.
- Read DJ. 1991.** Mycorrhizas in ecosystems. *Experientia* **47**(4): 376-391.
- Read DJ, Perez-Moreno J. 2003.** Mycorrhizas and nutrient cycling in ecosystems – a journey towards relevance? *New Phytologist* **157**(3): 475-492.

- Schweiger PF, Robson AD, Barrow NJ. 1995.** Root hair length determines beneficial effect of a *Glomus* species on shoot growth of some pasture species. *New Phytologist* **131**(2): 247-254.
- Valverde-Barrantes OJ, Freschet GT, Roumet C, Blackwood CB. 2017.** A worldview of root traits: the influence of ancestry, growth form, climate and mycorrhizal association on the functional trait variation of fine-root tissues in seed plants. *New Phytologist* **215**(4): 1562-1573.
- Valverde-Barrantes OJ, Horning AL, Smemo KA, Blackwood CB. 2016.** Phylogenetically structured traits in root systems influence arbuscular mycorrhizal colonization in woody angiosperms. *Plant and Soil* **404**(1-2): 1-12.
- Valverde-Barrantes OJ, Smemo KA, Blackwood CB, Norden N. 2015.** Fine root morphology is phylogenetically structured, but nitrogen is related to the plant economics spectrum in temperate trees. *Functional Ecology* **29**(6): 796-807.
- Valverde-Barrantes OJ, Smemo KA, Feinstein LM, Kershner MW, Blackwood CB. 2018.** Patterns in spatial distribution and root trait syndromes for ecto and arbuscular mycorrhizal temperate trees in a mixed broadleaf forest. *Oecologia* **186**(3): 731-741.
- Weemstra M, Mommer L, Visser EJ, van Ruijven J, Kuyper TW, Mohren GM, Sterck FJ. 2016.** Towards a multidimensional root trait framework: a tree root review. *New Phytologist* **211**: 1159–1169.
- Withington JM, Reich PB, Oleksyn J, Eissenstat DM. 2006.** Comparisons of structure and life span in roots and leaves among temperate trees. *Ecological Monographs* **76**(3): 381-397.
- Xu Y. 2011.** *Fine root morphology, anatomy and tissue nitrogen and carbon of the first five order roots in twenty seven Chinese tropical hardwood tree species.* Master, Northeast Forestry University, China.

Reviewer #2 (Remarks to the Author):

Thanks for this thorough revision of the manuscript. I find the answers to the reviewer requests well supported and overall convincing. I am also content with the revision of the text itself which is now much clearer in reasoning and more consistent in wording and line of argument in most cases. There is still one new part where the SRL is discussed which I find particularly hard to follow. I very much like the new paragraph discussing the absence of a non-linear relationship in non-woody species.

Just a few very minor comments:

L 143: I am aware that tToS was defined in the introduction but it is just not a very intuitive abbreviation. I think it would help to define it again in each sections (intro, results, discussion) at first mentioning.

L191: refer to table 1 again at the end of this sentence.

L193 ff: Here again I would find it easier to remind the reader of what you refer to as correlated and uncorrelated. Also use both terms in the figure caption of figure 4 please.

L 195: a mixed model or mixed models?

L 217: delete s from nutrients

L221: here Fig 2d would better serve

L277: Even though I like this additional paragraph explaining the relationship with SRL I am afraid it is not straight forward to follow through. Even after reading this part several times I cannot follow your line of argument relating SRL to your findings.

You state that: If RTD is positively correlated with root diameter, as predicted by the RES, SRL then mathematically scales negatively with RTD.

I think it is clear till that point. Now you start taking apart the SRL to RTD relationship for the two different regions of the nonlinear relationship. You argue:

However, for the region of the nonlinear curve where RTD slowly decreases with root diameter (Fig. 2c), SRL could be positively related to RTD.

To explain this last sentence you state:

This is because with increasing root diameter, the negative effect of root diameter (on SRL) could counteract the positive effect of RTD on SRL.

This is not really an explanation for me. You just state different relationships without background as to why the one or the other would be more likely related to biological background. Overall this makes a somewhat confusing and partly circular argument.

Sorry – I only now have the supplementary with the paper and see that figure 7 there does support this argument much better. However, as the figure is in the supplement, I think it is important for this discussion to be understandable without the supporting figure, so please try to make the logic of your argument more explicit and easier to follow.

L 291: Shouldn't it be RTD decreases and RN increases unless I misunderstand the sentence here. Perhaps better put this in two sentences to make it clearer.

L310: Does figure 6 show only non-woody species? Even if so that would not give a comparison to the woody ones directly. Of course you indicate the studies to the figure but I think it is too much to ask the reader checking what species types were in those studies. So please indicate in the figure legend.

Figure 4: in all other graphs you identify the different labels in the legend, only here you don't. I would suggest to be consistent with this and name correlated and non-correlated studies in the legend.

Supplementary:

Figure 5: I would delete the line as there is no significant relationship for figure b

Responses to reviewers' comments for version NCOMMS-18-21998A

Note, line numbers refer to our final Word version (NCOMMS-18-21998B) including 'track changes'.

REVIEWERS' COMMENTS:

Report from Reviewer #1

The resubmitted manuscript by Kong et al. on the non-linearity of root trait relationships has been much improved. Most comments have been well addressed and I appreciate the extra effort put into revising this manuscript; the text and writing is much more clear and straightforward. I still have some (minor) textual suggestions that the authors could consider to further clarify their interesting work before publication; in fact, sometimes the explanation in the rebuttal was more clear than in the manuscript. These mainly refer to the changed/added sections to this revised manuscript. I hope these are constructive.

Despite these (mostly textual) issues, I support publication of this manuscript in Nature Communications because of the novel, interesting and highly relevant insights in root trait relationships it demonstrates across a large species set.

Yours sincerely,
Monique Weemstra

L. 79 and throughout the manuscript: I do not propose the term 'uncorrelated trait relationships'; the fact that they are uncorrelated indicates that there are no relationships. I would replace this by a more correct or appropriate term, e.g. uncorrelated traits.

RESPONSE: Thanks for the suggestion. We now use "uncorrelated traits" throughout the manuscript.

I think the term 'correlated trait relationships' (L. 408-410), needs to be redefined. The definition suggests that the studies that reported correlated relationships do not include those that reported opposite correlations (i.e. in line with the RES), because these do not meet the definition, but because they had no significant effect on the non-linear relationships they were included (L. 418-420).

RESPONSE: In the text, we explain that studies reporting 'correlated trait relationships' included those studies reporting correlated trait relationships not supporting the RES (i.e.,

negative RTD-root diameter and positive RN-root diameter correlations) as well as some studies reporting trait relationships only partially supporting the RES (e.g., both positive RTD-root diameter and RN-root diameter relationships) (Lines 431-433, Lines 443-444).

Moreover, I really like how the data set is clarified by the new Suppl. Table 1.

RESPONSE: Thanks!

L. 83: replace ; by a comma and ‘that’ by ‘which’.

RESPONSE: Done.

L. 109-110: I appreciate that the authors have considered my earlier suggestion to check the effects of the hyphal mantle on RTD, but these seemed to be minor and thus irrelevant to include here; I would remove ‘As the mantle ... with RTD’.

RESPONSE: Thanks for this suggestion. If possible, we’d like to keep this sentence as it supports our argument that high root tissue density in EM species could arise mainly from change of root tissues rather than from the effects of the hyphal mantle. As the sentence was rather long, we have cut it into three sentences and amended it in response to the reviewer’s comment on L. 112:

“The mantle hyphae in EM species usually have low tissue density and have little correlation with RTD¹⁶, and therefore cannot not explain the observed differences between EM and AM species. Instead, EM species typically dominate in nutrient-poor soils²⁸, which may lead to thicker and/or more intensely lignified root cell walls³³. This, in turn, could potentially explain the higher RTD and lower RN in EM than in AM species^{15,17,30,31}.” (Lines 90-98)

L. 112: The authors aim to discuss the possibility of different root trait relationships between EcM and AM plant species, but I think this still needs some rephrasing/reorganizing to make it more clear. Based on the rebuttal, the authors suggest that poor soils (where EcM species usually dominate) lead to thicker/more lignified cell walls, potentially leading to higher RTD and lower RN for EcM species. If so, please rewrite this more clearly and explicitly. For instance, first mention the impacts on root cell walls and then, their effects on RTD and RN (it is much more clear in the rebuttal). Please include the right references here; I could not find evidence for changes in cell walls with soil fertility in the studies referred to.

RESPONSE: We have rewritten the sentence as follows:

“The mantle hyphae in EM species usually have low tissue density and have little correlation with RTD¹⁶, and therefore cannot not explain the observed differences between EM and AM

species. Instead, EM species typically dominate in nutrient-poor soils²⁸, which may lead to thicker and/or more intensely lignified root cell walls³³. This, in turn, could potentially explain the higher RTD and lower RN in EM than in AM species^{15,17,30,31}.” (Lines 90-98)

We acknowledge that there are few studies reporting direct evidence for the changes of cell walls with soil fertility. Instead, the references we cited here provide some indirect evidence. We have rephrased the sentence to emphasize this: “ ... *may* lead to thicker and/or more intensely lignified root cell walls”. Specifically, the cited references show that:

1) Variation of stele cell walls is positively related to RTD:

Wahl and Ryser (2000) [Reference 53, $r=0.52$, $p=0.02$]

2) RTD is higher in less fertile soils:

Holdaway et al. (2011) [Reference 17, a field study across fertility gradients]

Kramer-Walter et al. (2016) [Reference 15, a field study across New Zealand]

Kramer-Walter and Laughlin (2017) [Reference 31, a greenhouse experiment]

3) RTD is higher in EM species than in AM species:

Valverde-Barrantes et al. (2018) [Reference 30]

Kramer-Walter et al. (2016) [Reference 15]

L. 221: I think Fig. 2b should be 2d?

RESPONSE: Sorry for this mistake. Yes, it should be Fig. 2d. We have corrected it.

Fig. 2d: blue dots are missing in the legend

RESPONSE: The blue dots have been added.

L. 230-232: I would consider omitting this part. It sounds very obvious, and the key message is clear (and very interesting) without repeating these results: sampling bias matters for testing root trait relationships.

RESPONSE: Thanks for this suggestion. If possible, we'd like to keep this sentence because it provides a good-in lead for the following sentence.

L. 234: I think 'suggests' can be replaced by 'demonstrates' as the authors refer to the patterns they show in Fig. 2.

RESPONSE: Done.

L. 223-226: I appreciate the reformulation, but it remains a bit vague. Is it necessary to include the AM species here (L. 225-226) – if not, please remove. Why not simply say something like ‘For EM species, thin roots are covered by (a thicker?) EM fungal mantle, which is relatively rich in N; this thus enhances the root N concentrations of thin roots compared to thick roots’. I am not a native English speaker, so please correct his where needed, but I would keep it more simple and explicit.

RESPONSE: Thanks for this suggestion. Following the reviewer’s suggestion, we have much simplified this sentence. However, if possible, we’d like to keep the comparison between EM and AM species as it makes such an interesting contrast.

“First, for EM species, thin absorptive roots are covered by a relatively thick EM fungal mantle¹⁶, which is relatively rich in N³⁸; this thus enhances the root N concentration of thin roots compared to thick roots. This is notably different from AM species where thicker absorptive roots are usually associated with greater mycorrhizal colonization^{18,24,39}.” (Lines 228-232)

L. 247: perhaps replace ‘applied to’ by ‘as observed on leaves’, to be more specific?

RESPONSE: We have replaced ‘as applied to’ by ‘as observed for’. However, we’d like to keep ‘aboveground plant organs’ instead of ‘leaves’ because the references cited here focus on the aboveground plant organs, including leaves *and* stems.

L. 278: This argument is not entirely clear, probably because there is no positive effect of RTD on SRL. Should this not be replaced by ‘a potential positive effect of RTD on SRL’? This would imply that there may be a positive effect of RTD on SRL, but it is not visible due to the strong negative effect of diameter on RTD for thick roots. Is this what the authors mean?

RESPONSE: Thanks for this suggestion. The reviewer’s understanding is overall consistent with our argumentation. We have accepted the term of ‘a potential positive effect of RTD on SRL’ and revised the sentence as follows:

“...the negative effect of root diameter on SRL could counteract a potential positive effect of RTD on SRL, which, in turn, would lead to a positive relationship of SRL with RTD.” (Lines 287-289)

L. 287: Replace ‘does not conform to’ by ‘does not necessarily agree with’ or so.

RESPONSE: Done.

L. 303 – 321: This explanation for the different trait relationships for woody and non-woody species is not very clear. I hope the authors can explain the following a bit better:

- L. 309-311: It is suggested that non-woody species change root hairs and branching to acquire resources rather than investing in mycorrhiza compared to woody species. However, woody species also show lower colonization rates on more fertile soils, so I am not sure if this explains the differences between the growth forms (*Question 1*). Moreover, the manuscript does not clarify how this is related to a reduction in the RTD of non-woody species (L. 310); perhaps a part of the rebuttal that explains the link between cortex size (and thus RTD) and hairs and branching could be briefly incorporated (*Question 2*)?

RESPONSE: Thanks for these insight comments and suggestions!

Fig. 1c from Ma et al. (2018). Relationships between mycorrhizal colonization and root diameter in woody (brown circles and line) and herbaceous (green circles and line) species.

As for *Question 1*, we note from Ma’s study (see the above figure) which showed a constant lower mycorrhizal colonization in non-woody than in woody species across a wide range root diameters (slopes are equal; intercept is lower in non-woody than in woody species). As roots in this study were sampled from a wide variety of ecosystems (subtropical forests, temperate forests, grasslands) across a large gradient of soil fertility (Wang et al. 2018), it is likely that the lower mycorrhizal colonization in non-woody compared to woody species holds true for both fertile and infertile soils.

As for *Question 2*, we have taken the reviewer’s advice and added part of the text from our first rebuttal letter regarding the cortex size:

“Compared with absorptive roots of woody species at a given diameter, non-woody species are reported to have about 30% less mycorrhizal colonization¹² while having a higher proportion of root cortex (t-test, $p < 0.01$) and thus lower RTD. This higher proportion of

cortex in non-woody species might be associated with foraging strategies other than mycorrhizal colonization (e.g., metabolic activity). Therefore, in fertile soils, absorptive roots of non-woody species may be less dense and more active than absorptive roots of woody species⁵³ (Supplementary Fig. 6), and have more root hairs⁵⁴ and/or root branching⁵⁵. Together, this might offset the lower nutrient acquisition through mycorrhizal associations in non-woody species.” (Lines 322-330)

• L.313-316: Does a higher RTD also reduce mycorrhizal colonization for EcM host species that may be less related to cortex size? The studies referred only include few (Kong et al. 2014, their Fig. 3) or no (Sharda & Koide 2008) EcM host species.

RESPONSE: We have little direct evidence for the argument of reduced mycorrhizal colonization with higher RTD. However, some studies provide indirect evidence supporting the argument. We have cited these studies in the revised version:

Brundrett (2002): EcM colonization of some EcM gymnosperms is blocked by cortex with thickened cell walls (see page 292 in Brundrett’s paper).

Withington et al. (2006): EcM angiosperms with higher RTD have lower proportion of root cross-sectional area colonized by the EcM fungi.

Kramer et al. (2016): EcM species have higher RTD in infertile soils.

• L. 315-316: I am not sure this is relevant. It is a bit confusing and has already been mentioned (L. 308), so please consider removing this.

RESPONSE: We have removed this sentence.

• L. 316-317: Why is stele vessel size relevant in this explanation?

RESPONSE: Stele vessel size is not particularly relevant to the explanation and we have deleted it.

• L. 317-318: The wording in this sentence seems incorrect, please rephrase. Also, I am not sure that the results of Ma et al. 2017 imply that woody species are always in greater need of mycorrhizal symbiosis than non-woody species on both fertile and infertile soils.

RESPONSE: Thanks for this comment. We have revised this sentence as follows:

“However, in infertile soils, roots of non-woody species could be denser⁵³ relative to woody species because of lower dependence on mycorrhizal colonization for non-woody than for

woody species¹².” (Lines 333-336)

Fig. 1c from Ma et al. 2018 Relationships between mycorrhizal colonization and root diameter in woody (brown circles and line) and herbaceous (green circles and line) species.

As for the second point, Ma et al. (2018) showed constantly lower mycorrhizal colonization in non-woody than in woody species across a wide range of root diameters (see figure above). As roots in this study were sampled from a wide variety of ecosystems (subtropical forests, temperate forests, grasslands) across a large gradient of soil fertility (Wang et al. 2018), it is likely that the lower mycorrhizal colonization in non-woody compared to woody species holds true for both fertile and infertile soils.

• L. 318-320: ‘These ...’ speculations/findings...? Also, roots of non-woody species do not necessarily have greater variation in RTD and root N than woody species for a given diameter based on Fig. 2.

RESPONSE: We have revised the sentence by adding ‘speculations’, removing ‘for a given diameter’, and combining this sentence and the next sentence into one sentence:

“These speculations could partially explain why roots of non-woody species show greater variation of RTD and RN (and hence weak nonlinear root trait relationships) compared to woody species.” (Lines 337-339)

• L. 320: Based on the above issues, I am afraid I don’t see how the authors explain the lack of non-linear trait relationships for non-woody species.

RESPONSE: We appreciate the reviewer’s valuable comments and suggestions regarding the differences between woody and non-woody species. By incorporating these suggestions, and adding some additional explanation and clarification in this final version of the manuscript,

we hope our reasoning is clear now.

L. 324: Should 'However' be 'In contrast'?

RESPONSE: Done.

Reviewer #2 (Remarks to the Author):

Thanks for this thorough revision of the manuscript. I find the answers to the reviewer requests well supported and overall convincing. I am also content with the revision of the text itself which is now much clearer in reasoning and more consistent in wording and line of argument in most cases. There is still one new part where the SRL is discussed which I find particularly hard to follow. I very much like the new paragraph discussing the absence of a non-linear relationship in non-woody species.

RESPONSE: We appreciate the reviewer's comments and our detailed responses to the remaining comments are listed below.

Just a few very minor comments:

L 143: I am aware that tToS was defined in the introduction but it is just not a very intuitive abbreviation. I think it would help to define it again in each sections (intro, results, discussion) at first mentioning.

RESPONSE: Accepted.

L191: refer to table 1 again at the end of this sentence.

RESPONSE: Done.

L193 ff: Here again I would find it easier to remind the reader of what you refer to as correlated and uncorrelated. Also use both terms in the figure caption of figure 4 please.

RESPONSE: Thanks for this suggestion. We have amended the sentence as follows:

“For woody species, the nonlinear relationships between RTD and root diameter and between RN and root diameter were better fitted for studies (i.e., data sources) reporting correlated trait relationships (i.e., negative RTD-root diameter and positive RN-root diameter correlations) ($R^2=0.30$ and $R^2=0.049$) than for studies reporting uncorrelated traits ($R^2=0.06$ and $R^2=0.026$) (Fig. 4; Supplementary Data 2).” (Lines 196-201)

We have also added both terms to the legend of Figure 4 (Lines 683-687).

L 195: a mixed model or mixed models?

RESPONSE: It should be a mixed model. We have revised it accordingly.

L 217: delete s from nutrients

RESPONSE: Done.

L221: here Fig 2d would better serve

RESPONSE: Sorry for this mistake. It should be Fig. 2d and we have corrected it.

L277: Even though I like this additional paragraph explaining the relationship with SRL I am afraid it is not straight forward to follow through. Even after reading this part several times I cannot follow your line of argument relating SRL to your findings.

You state that: If RTD is positively correlated with root diameter, as predicted by the RES, SRL then mathematically scales negatively with RTD.

I think it is clear till that point. Now you start taking apart the SRL to RTD relationship for the two different regions of the nonlinear relationship. You argue:

However, for the region of the nonlinear curve where RTD slowly decreases with root diameter (Fig. 2c), SRL could be positively related to RTD.

To explain this last sentence you state:

This is because with increasing root diameter, the negative effect of root diameter (on SRL) could counteract the positive effect of RTD on SRL.

This is not really an explanation for me. You just state different relationships without background as to why the one or the other would be more likely related to biological background. Overall this makes a somewhat confusing and partly circular argument.

RESPONSE: Thanks for these thoughts. We are sorry for any confusion we may have caused.

We will clarify these points below:

(1) Our reasoning is based on the nonlinear and negative relationship between RTD and root diameter. Under this nonlinear relationship, the two components of SRL (i.e., RTD and root diameter, Ryser 2006) have opposing effects on SRL according to the following mathematical relationship (Ryser 2006; Holdaway et al. 2011). There are two reasons for why negative effects of root diameter could counteract potential positive effects of RTD for the region of the nonlinear curve with a slow decrease of RTD with increasing root diameter. 1) A smaller positive effect of RTD relative to the larger negative effect of root diameter on SRL with increasing root diameter. 2) The negative effect of root diameter on SRL would be magnified as root diameter is in second power whereas RTD is in first power in the denominator of the above formula.

(2) Our argumentation for the relationship of SRL with other root traits is based on the nonlinear relationship between RTD and root diameter. As we demonstrate, such nonlinear relationship is derived from the allometric relationship between root stele and cortex tissues. Our argumentation in this section concentrates on the region of the nonlinear curve with a slow decrease of RTD with increasing root diameter (see Fig. 2c). Here, the slow decrease of RTD can be attributed to the slow decrease of the proportion of root cross-sectional area accounted for by root stele (Fig. 1b). We argue that less stele tissue or more cortex tissue with increasing root diameter might result from stronger dependence of nutrient foraging on mycorrhizal fungi for thicker absorptive roots (Baylis 1975; Liu et al. 2015; Eissenstat et al. 2015). We mention this biological explanation in Lines 270-279.

(3) Our argumentation is not circular. In this section of our manuscript, we try to understand the relationships of SRL with RTD and RN, as shown in previous studies, according to two trait relationships, i.e., our finding of the nonlinear relationship between RTD and root diameter, and the mathematical relationship of SRL with RTD and root diameter: $4/(\pi \times \text{RTD} \times \text{root diameter}^2)$. These two trait relationships are essentially different from each other because 1) the formula of $\text{SRL} = 4/(\pi \times \text{RTD} \times \text{root diameter}^2)$ is derived from the definition of SRL and RTD and the assumption that roots are cylindrically shaped; 2) the nonlinear relationship between RTD and root diameter is derived from the allometric relationship between root stele and cortex tissues, and is based on some biophysical mechanisms (see Kong et al. 2017). This means that the formula of $\text{SRL} = 4/(\pi \times \text{RTD} \times \text{root diameter}^2)$ holds regardless of the nonlinear relationship between RTD and root diameter, and *vice versa*. Therefore, there should be no circular argument for our discussion in this section.

We have simplified the explanation in the main text:

“However, for the region of the nonlinear curve where RTD slowly decreases with root diameter (Fig. 2c), the negative effect of root diameter on SRL could counteract a potential positive effect of RTD on SRL, which, in turn, would lead to a positive relationship of SRL with RTD. In contrast, for the region of the nonlinear curve with fast decrease of RTD (Fig. 2c), SRL may show no relationship with RTD. This is because with increasing root diameter, the negative effect of root diameter on SRL could be offset by the potential positive effect of RTD on SRL.” (Lines 286-293)

Sorry – I only now have the supplementary with the paper and see that figure 7 there does support this argument much better. However, as the figure is in the supplement, I think it is

important for this discussion to be understandable without the supporting figure, so please try to make the logic of your argument more explicit and easier to follow.

RESPONSE: We initially explored whether the poor relationships of SRL with RTD and RN (i.e., Supplementary Figure 7a,b), as shown in previous studies, could be explained by our finding of allometry-based nonlinearity combined with the formula of $SRL = 4/(\pi \times RTD \times \text{root diameter}^2)$. Here, we try to make this logic more explicit and easier to follow:

1) We speculate that RTD would show different relationships with SRL for the region of the nonlinear curve with slow decrease and for the region of the curve with fast decrease of RTD with increasing root diameter. This could lead to a weak relationship between RTD and SRL across the whole range of nonlinear curve as shown in Supplementary Figure 7a.

2) RTD is usually negatively correlated with RN because higher RTD will be accompanied with greater investment of nitrogen in the cell wall fraction (see Supplementary Figure 6), thus leading to lower root activity and RN. Given the weak relationship between RTD and SRL, SRL would also be weakly correlated with RN (Supplementary Figure 7b).

3) The weak correlation between SRL and RN could also contribute to the weak correlation between root diameter and RN (see Fig. 2 in the main text) because SRL is strongly coupled with root diameter (Supplementary Figure 7b).

We have revised the relevant sentences as follows:

“Together, the nonlinear relationship between RTD and root diameter could explain an overall weak correlation of SRL with RTD, and also with RN (Supplementary Fig. 7a,b)^{11,14,15,49} across the whole region of the nonlinear curve. Moreover, the weak correlation between SRL and RN and the strong coupling of SRL with root diameter^{12,14,15,50} (also see Supplementary Fig.7c) could also explain the relative weak correlation between root diameter and RN (Fig. 2).” (Lines 293-300).

L 291: Shouldn't it be RTD decreases and RN increases unless I misunderstand the sentence here. Perhaps better put this in two sentences to make it clearer.

RESPONSE: Our apologies, the reviewer is correct. We have corrected this mistake.

L310: Does figure 6 show only non-woody species? Even if so that would not give a comparison to the woody ones directly. Of course you indicate the studies to the figure but I think it is too much to ask the reader checking what species types were in those studies. So

please indicate in the figure legend.

RESPONSE: Most species in this figure are woody, except one non-woody species in Kong et al. (2016), i.e., reference 13 in the Supplementary References. We have now added this detail to the figure legend.

Figure 4: in all other graphs you identify the different labels in the legend, only here you don't. I would suggest to be consistent with this and name correlated and non-correlated studies in the legend.

RESPONSE: We have added the labels to the figure panels:

Supplementary:

Figure 5: I would delete the line as there is no significant relationship for figure b

RESPONSE: Done.

References cited in this response letter:

Baylis GTS 1975. The magnolioid mycorrhiza and mycotrophy in root systems derived from it. In: Sanders FE, Mosse B, Tinker PB eds. *Endomycorrhizas*. London: Academic Press, 373–389.

Brundrett MC. 2002. Coevolution of roots and mycorrhizas of land plants. *New Phytologist* **154**(2): 275-304.

Eissenstat DM, Kucharski JM, Zadworny M, Adams TS, Koide RT. 2015. Linking root traits to nutrient foraging in arbuscular mycorrhizal trees in a temperate forest. *New Phytologist* **208**(1): 114-124.

Holdaway RJ, Richardson SJ, Dickie IA, Peltzer DA, Coomes DA. 2011. Species- and community-level patterns in fine root traits along a 120 000-year soil chronosequence in temperate rain forest. *Journal of Ecology* **99**(4): 954-963.

- Kramer-Walter KR, Bellingham PJ, Millar TR, Smissen RD, Richardson SJ, Laughlin DC, Mommer L. 2016.** Root traits are multidimensional: specific root length is independent from root tissue density and the plant economic spectrum. *Journal of Ecology* **104**(5): 1299-1310.
- Kramer-Walter KR, Laughlin DC. 2017.** Root nutrient concentration and biomass allocation are more plastic than morphological traits in response to nutrient limitation. *Plant and Soil*(416): 539-550.
- Liu B, Li H, Zhu B, Koide RT, Eissenstat DM, Guo D. 2015.** Complementarity in nutrient foraging strategies of absorptive fine roots and arbuscular mycorrhizal fungi across 14 coexisting subtropical tree species. *New Phytologist* **208**(1): 125-136.
- Ma Z, Guo D, Xu X, Lu M, Bardgett RD, Eissenstat DM, McCormack ML, Hedin LO. 2018.** Evolutionary history resolves global organization of root functional traits. *Nature* **555**(7694): 94-97.
- Ryser P. 2006.** The mysterious root length. *Plant and Soil* **286**(1-2): 1-6.
- Valverde-Barrantes OJ, Smemo KA, Feinstein LM, Kershner MW, Blackwood CB. 2018.** Patterns in spatial distribution and root trait syndromes for ecto and arbuscular mycorrhizal temperate trees in a mixed broadleaf forest. *Oecologia* **186**(3): 731-741.
- Wahl S, Ryser P. 2000.** Root tissue structure is linked to ecological strategies of grasses. *New Phytologist* **148**(3): 459-471.
- Wang R, Wang Q, Zhao N, Xu Z, Zhu X, Jiao C, Yu G, He N, Niu S. 2018.** Different phylogenetic and environmental controls of first-order root morphological and nutrient traits: Evidence of multidimensional root traits. *Functional Ecology* **32**(1): 29-39.
- Withington JM, Reich PB, Oleksyn J, Eissenstat DM. 2006.** Comparisons of structure and life span in roots and leaves among temperate trees. *Ecological Monographs* **76**(3): 381-397.

Reviewer #1 (Remarks to the Author):

I am sorry to see that one of the main relationships (RN – diameter) on which this study is based turns out to be not significant. However, the manuscript focuses mostly on the relationship between RTD and diameter though, which was the strongest relationship to begin with, and of the different relationships between RN and diameter within the woody species that still are significant. Therefore, I think the manuscript requires only few adjustments:

1. L. 144-146 should be rephrased, because the R², P-value and the data (Fig. 2b) indicate no relationship between root N and diameter across all species.
2. Because the relationship between RN and diameter within the woody species becomes more relevant now (as this relationship was not significant across all species), it needs to be more clearly illustrated and explained, especially in the Results section:
 - a. The discussion but not the results mentions that this relationship was negative for EM species and positive for AM and ErM woody species (L 213-214). This needs to be added to the results section too, e.g. after mentioning the relationships between RTD-diameter for all mycorrhizal groups (L. 158-162).
 - b. L. 165-167 are unclear now: the previous line suggests that these results refer to the PIC (L 162-165), but the reference to Fig. 2d (L. 167) suggests it does not. So, please clarify this.
 - c. Unless I overlooked it, the direction (positive/negative) of these relationships cannot -be found anywhere in the manuscript: neither the statistics nor the data in Fig. 2d show clearly positive or negative non-linear relationships. I urge to add regression lines to Fig. 2 for any significant relationship.
 - d. Do these contrasting relationships account for the non-significant relationship across all species? This could for instance be discussed at paragraph L. 213-222.
3. In the conclusion (L. 328-331), it would be valuable to add that these important non-linear relationships may be different for EM species, which is a large group of species and which showed opposite patterns in one of the 2 main relationships tested here. The authors have discussed a possible explanation for these opposite patterns (L. 213-222) but it would be relevant to also mention their implications in terms of revealing insights in the drivers of root structure (L. 327-331).
4. Finally, I urge the authors to refer to specific supplementary information (i.e. tables or figures,) rather than e.g. Supplementary data 2. For instance, in L. 167, I do not see which supplementary data the authors refer to exactly. Suppl. Fig. 2 does not show the statistics, and Suppl. Table 2 includes non-woody species only.

I hope these comments are clear and useful.

Yours sincerely,

Monique Weemstra

Reviewer #2 (Remarks to the Author):

Nature Communications NCOMMS-18-21998B

Corresponding Author: Deliang Kong

Thanks for this second revision of the manuscript. I have thoroughly read this version and found it again much improved and more concise and convincing in argument. I am content that the points raised by both reviewers have been addressed and solved in this version of the manuscript and I am content with the answers given to my comments on the first revision. I find this version ready for publication.

Very minor points:

L 87: cannot not explain?

L 108: EM plant growing – something missing here?

L126 space missing after 0.068

Note, line numbers refer to our final Word version (NCOMMS-18-21998C) including 'track changes'.

REVIEWERS' COMMENTS:

Reviewer #1 (Remarks to the Author):

I am sorry to see that one of the main relationships (RN – diameter) on which this study is based turns out to be not significant. However, the manuscript focuses mostly on the relationship between RTD and diameter though, which was the strongest relationship to begin with, and of the different relationships between RN and diameter within the woody species that still are significant. Therefore, I think the manuscript requires only few adjustments:

1. L. 144-146 should be rephrased, because the R^2 , P-value and the data (Fig. 2b) indicate no relationship between root N and diameter across all species.

RESPONSE: We have revised this sentence as follows:

“...and the nonlinear relationship between RN and root diameter was rather weak and not significant (nonlinear regression, $R^2=0.002$, $p>0.1$) (Fig. 2a, b, Supplementary Data 2).” (Lines 145-148)

2. Because the relationship between RN and diameter within the woody species becomes more relevant now (as this relationship was not significant across all species), it needs to be more clearly illustrated and explained, especially in the Results section:

a. The discussion but not the results mentions that this relationship was negative for EM species and positive for AM and ErM woody species (L 213-214). This needs to be added to the results section too, e.g. after mentioning the relationships between RTD-diameter for all mycorrhizal groups (L. 158-162).

RESPONSE: We have added new text on the RN-root diameter relationship for different mycorrhizal groups to the Results section:

“...The relationship between RN and root diameter was negative in EM woody species (nonlinear regression, $R^2=0.073$, $p<0.001$) and positive in AM (nonlinear regression, $R^2=0.02$, $p<0.001$) and ERM (nonlinear regression, $R^2=0.28$, $p<0.001$) woody species (Fig. 2d; regression equations are presented in Supplementary Data 2).”

(Lines 165-169)

b. L. 165-167 are unclear now: the previous line suggests that these results refer to the PIC (L. 162-165), but the reference to Fig. 2d (L. 167) suggests it does not. So, please clarify this.

RESPONSE: To clarify this point, we have rewritten L. 165-167 and moved it after mentioning the relationships between RTD-diameter for all mycorrhizal groups (L. 158-162) as in the reviewer's above comment (i.e., comment 2a):

"...The relationship between RN and root diameter was negative in EM woody species (nonlinear regression, $R^2=0.073$, $p<0.001$) and positive in AM (nonlinear regression, $R^2=0.02$, $p<0.001$) and ERM (nonlinear regression, $R^2=0.28$, $p<0.001$) woody species (Fig. 2d; regression equations are presented in Supplementary Data 2)."
(Lines 165-169)

c. Unless I overlooked it, the direction (positive/negative) of these relationships cannot -be found anywhere in the manuscript: neither the statistics nor the data in Fig. 2d show clearly positive or negative non-linear relationships. I urge to add regression lines to Fig. 2 for any significant relationship.

RESPONSE: To clarify the direction of the relationships, we have followed the reviewer's suggestion and added the regression lines to Fig. 2. Further, the direction of the relationships in Fig. 2d can also be learnt from the regression equations as presented in Supplementary Data 2. For example, the relationship for EM woody species is negative given the regression equation: $y=(3.46+0.14 x^{-1})^2$, where y is RN, and x is root diameter.

d. Do these contrasting relationships account for the non-significant relationship across all species? This could for instance be discussed at paragraph L. 213-222.

RESPONSE: The non-significant relationship refers to the RN-root diameter relationship for all species, i.e., woody species + non-woody species. However, the relationship is significant for woody species and for different mycorrhizal groups within the woody species. In fact, paragraph L. 213-222 addresses the different RN-root diameter relationships among mycorrhizal groups only in woody species. To clarify this, we have rephrased the first sentence of this paragraph as follows:

"Interestingly, for woody species we found a negative relationship between RN and root diameter in EM species, while the relationship was positive in AM and ERM

species (Fig. 2d).” (Lines 221-223)

3. In the conclusion (L. 328-331), it would be valuable to add that these important non-linear relationships may be different for EM species, which is a large group of species and which showed opposite patterns in one of the 2 main relationships tested here. The authors have discussed a possible explanation for these opposite patterns (L. 213-222) but it would be relevant to also mention their implications in terms of revealing insights in the drivers of root structure (L. 327-331).

RESPONSE: Thanks for the suggestion. We have added a sentence incorporating these two points:

“Interestingly, EM species show a different RN-root diameter relationship from that found in AM and ERM species, probably because EM species have a thinner fungal mantle and less hyphal foraging precision in thicker absorptive roots.” (Lines 339-341)

4. Finally, I urge the authors to refer to specific supplementary information (i.e. tables or figures,) rather than e.g. Supplementary data 2. For instance, in L. 167, I do not see which supplementary data the authors refer to exactly. Suppl. Fig. 2 does not show the statistics, and Suppl. Table 2 includes non-woody species only.

RESPONSE: We kindly point the fact that Line 167 referred to Fig. 2d, not Fig. S2 or Table S2. However, we have checked throughout the manuscript that references to the supplementary information are as specific as possible. For example, in L. 167 (i.e., Lines 168-169 in the submitted version with track changes), we have added ‘Fig. 2d; regression equations are presented in Supplementary Data 2’.

We have also added the statistics to Supplementary Fig. 2 and also to Supplementary Fig. 3.

There is only one sentence referring to Supplementary Table 2 in the manuscript (i.e., Lines 156-159) which shows the results only for non-woody species. To specify the species pool which Supplementary Table 2 refers to, we have moved ‘for non-woody species’ in Line 157 just before ‘(Supplementary Table 2)’ in this sentence:

“When considered separately, the relationships of RTD and RN with root diameter were weak (Figs. 2e, f, Supplementary Fig. 1, 2; regression equations are presented in Supplementary Data 2) and unaffected by mycorrhizal type for non-woody species

(Supplementary Table 2).” (Lines 165-169)

I hope these comments are clear and useful.

RESPONSE: We greatly appreciate these comments and suggestions which are much clear and helpful for improvement of our manuscript.

Yours sincerely,
Monique Weemstra

Reviewer #2 (Remarks to the Author):

Nature Communications NCOMMS-18-21998B

Corresponding Author: Deliang Kong

Thanks for this second revision of the manuscript. I have thoroughly read this version and found it again much improved and more concise and convincing in argument. I am content that the points raised by both reviewers have been addressed and solved in this version of the manuscript and I am content with the answers given to my comments on the first revision. I find this version ready for publication.

RESPONSE: Thanks for these positive comments.

Very minor points:

L 87: cannot not explain?

RESPONSE: Sorry for this mistake. It should be “cannot explain”.

L 108: EM plant growing – something missing here?

RESPONSE: It should be “where EM plants grow”. We have revised it. (Line 109)

L126 space missing after 0.068

RESPONSE: Done.